# Rule Based Learning with Dynamic (Graph) Neural Networks

## Abstract

A common problem of classical neural network architectures is that additional information or expert knowledge cannot be naturally integrated into the learning process. To overcome this limitation, we propose a two-step approach consisting of (1) generating formal rules from knowledge and (2) using these rules to define rule based layers – a new type of dynamic neural network layer. The focus of this work is on the second step, i.e., rule based layers that are designed to dynamically arrange learnable parameters in the weight matrices and bias vectors for each input sample following a formal rule. Indeed, we prove that our approach generalizes classical feed-forward layers such as fully connected and convolutional layers by choosing appropriate rules. As a concrete application we present rule based graph neural networks (RuleGNNs) that are by definition permutation equivariant and able to handle graphs of arbitrary sizes. Our experiments show that RuleGNNs are comparable to state-of-the-art graph classifiers using simple rules based on the Weisfeiler-Leman labeling and pattern counting. Moreover, we introduce new synthetic benchmark graph datasets to show how to integrate expert knowledge into RuleGNNs making them more powerful than ordinary graph neural networks.

## 1 Introduction

Using expert knowledge to increase the efficiency, interpretability or predictive performance of a neural network is an evolving research direction in machine learning [21, 23]. Many ordinary neural network architectures are not capable of using external and structural information such as expert knowledge or meta-data, e.g., graph structures in a dynamic way. We would like to motivate the importance of "expert knowledge" by considering the following example. Maybe one of the best studied examples based on knowledge integration are convolutional neural networks [12]. Convolutional neural networks for images use at least two extra pieces of "expert knowledge" that is: *neighbored pixels correlate*, and *the structure of images is homogeneous*. The consequence of this *knowledge* is the use of receptive fields and weight sharing. It is a common fact that the usage of this information about images has highly improved the predictive performance over fully connected neural networks. But what if expert knowledge suggests that rectangular convolutional kernels are not suitable to solve the task? In this case the ordinary convolutional neural network architecture is too *static* to adapt to the new information. Dynamic neural networks are not only applicable to images but also to other data types such as video [25], text [10], or graphs [19]. The limitation of such approaches is that expert knowledge is somehow implicit and not directly encoded in the network structure, i.e., for each new information a new architecture has to be designed. Thus, our goal is to extract the essence of dynamic neural networks by defining a new type of neural network layer that is on the one side able to use expert knowledge in a dynamic way and on the other side easily configurable. Our solution to this problem are rule based layers that are able to encode expert

Submitted to 38th Conference on Neural Information Processing Systems (NeurIPS 2024). Do not distribute.

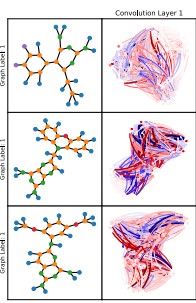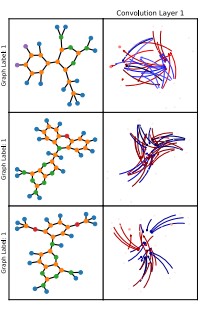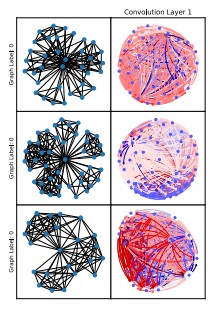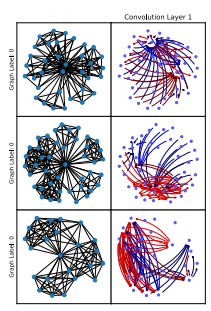

(a) Learned weights and bias for the best model of the DHFR dataset.

(b) Learned weights for the best model of the IMDB-BINARY dataset.

Figure 1: Visualization of the learnable parameters of our RuleGNN on DHFR (a) and IMDB-BINARY (b) for three different graphs. Positive weights are denoted by red arrows and negative weights by blue arrows. The arrow thicknesss and color corresponds to the absolute value of the weight. The bias is denoted by the size of the node. The second image of (a) resp. (b) shows the weights the 10 resp. 5 largest positive and negative weights.

knowledge directly in the network structure. As far as we know, this is the first work that defines a dynamic neural network layer in this generality.

**Main Idea**  We simplify and unify the integration of expert knowledge and additional information into neural networks by proposing a two-step approach and show how to encode given extra information directly into the structure of a neural network in a dynamic way. In the *first step* the extra information or expert knowledge is formalized using appropriate rules (e.g., *certain pixels in images are important*, *only nodes in a graph of type A and B interact*, *some patterns, e.g., cycles or cliques, in a graph are important*, etc.). In the *second step* the rules are used to manipulate the structure of the neural network. More precisely, the rules determine the positions of the weights in the weight matrix and the bias terms. We note that the focus of this work is on the second step as we show how to use given rules to dynamically adapt the layers. In fact, we do not provide a general instruction for deriving formal rules from given expert knowledge. In difference to ordinary network layers we consider a set $\mathcal{W}$ of learnable parameters instead of fixed weight matrices. The weight matrices and bias terms are then constructed for each input sample independently using the learnable parameters from $\mathcal{W}$. Indeed, each learnable parameter in $\mathcal{W}$ is associated with a specific relation between an input and output feature of a layer. As an example consider Figure 1 where each input and output feature corresponds to a specific node in the graph. The input samples are (a) molecule graphs respectively (b) snippets of social networks and the task is to predict the graph class. Each colored arrow in the figure corresponds to a learned parameter from $\mathcal{W}$, i.e., a specific relation between two atoms in the molecules or two nodes in the social network. Considering only the weights with the largest absolute values, see the second image of (a) respectively (b), our approach has learned how to propagate information from outer atoms to the rings respectively from the nodes to the "important" nodes of the social network. This example shows several advantages of our approach: (1) rule based layer type has a much more flexible structure than layers in classical architectures and allow to deal with *arbitrary input dimensions*, (2) the layers are easily integrable into existing architectures, and (3) the learned parameters, hence the model, is interpretable and can possibly be used to extract new knowledge from the data or to improve the existing rules.

**Main Contributions**  We define a new type of neural network layer called rule based layer. This new layer can be integrated into arbitrary architectures making them dynamic, i.e., the structure of the network changes based on the input data and predefined rules. We prove that rule based layers generalize classical feed-forward layers such as fully connected and convolutional layers. Additionally, we show that rule based layers can be applied to graph classification tasks, by introducing RuleGNNs, a new type of graph neural networks. In this way we are able to extend the concept of dynamic neural networks to graph neural networks together with all the advantages of dynamic neural networks, e.g., that RuleGNNs are by definition permutation equivariant and able to handle graphs

72 of arbitrary sizes. Considering various real-world graph datasets, we demonstrate that RuleGNNs
73 are competitive with state-of-the-art graph neural networks and other graph classification methods.
74 Using synthetic graph datasets we show that "expert knowledge" is easily integrable into our neural
75 network and also necessary for classification[1]

76 The rest of the paper is organized as follows: We introduce the concept of rule based layers in Section 2
77 and prove in Section 3 that rule based layers generalize fully connected and convolutional layers.
78 In Section 4 we present RuleGNNs and apply them in Section 5 to different benchmark datasets
79 and compare the results with state-of the art graph neural networks. Finally, we discuss limitations,
80 related work and conclude the paper in Section 6.

## 2  Rule Based Learning

82 Introducing the concept of rule based learning we first present some basic definitions followed by the
83 formal definition of rule based layers.

84 **Preliminaries**  For some $n \in \mathbb{N}$ we denote by $[n]$ the set $\{1, \ldots, n\}$. A neural network is denoted
85 by a function $\mathbf{f}(-, \Theta) : \mathbb{R}^n \longrightarrow \mathbb{R}^m$ with the learnable parameters $\Theta$. We extend this notation
86 introducing an additional parameter $\mathcal{R}$, that is the set of formal rules $\mathcal{R} = \{\mathbf{R}^1, \ldots, \mathbf{R}^k\}$. The
87 exact definition of these rules is given in the next paragraph. Informally, a rule $\mathbf{R}$ is a function
88 that determines the distribution of the weights in the weight matrix or the bias vector of a layer. A
89 rule $\mathbf{R}$ is called *dynamic* if it is a function in the input samples $x \in \mathbb{R}^n$ otherwise it is called *static*.
90 An example of a static rule is the one used to define convolutional layers, see Proposition 2. An
91 example of a dynamic rule can be found in Section 4. In our setting, a neural network is a function
92 $\mathbf{f}(-, \Theta, \mathcal{R}) : \mathbb{R}^* \longrightarrow \mathbb{R}^*$ that depends on a set of learnable parameters denoted by $\Theta$ and some
93 rule set $\mathcal{R}$ derived from expert knowledge or additional information. The notation $*$ in the domain
94 and codomain of $\mathbf{f}$ indicates that the input and output can be of arbitrary or variable dimension. As
95 usual $\mathbf{f}$ is a concatenation of sub-functions $f^1, \ldots, f^l$ called the layers of the neural network. More
96 precisely, the $i$-th layer is a function $f^i(-, \Theta^i, \mathbf{R}^i) : \mathbb{R}^* \longrightarrow \mathbb{R}^*$ where $\Theta^i$ is a subset of the learnable
97 parameters $\Theta$ and $\mathbf{R}^i$ is an element of the ruleset $\mathcal{R}$. We call a layer $f^i$ *static* if $\mathbf{R}^i$ is a static rule and
98 *dynamic* if $\mathbf{R}^i$ is a dynamic rule. The input data is a triple $(\mathbf{D}, \mathbf{L}, \mathbf{I})$, where $\mathbf{D} = \{x_1 \ldots, x_k\}$ with
99 $x_i \in \mathbb{R}^*$ is the set of examples drawn from some unknown distribution. The labels are denoted by
100 $\mathbf{L} = (y_1 \ldots, y_k)$ with $y_i \in \mathbb{R}^*$ and $\mathbf{I}$ is some additional information known about the input data $\mathbf{D}$.
101 This can be for example knowledge about the graph structure, node or edge labels, importance of
102 neighborhoods and many more. One main assumption of this paper is that $\mathbf{I}$ can be used to derive a
103 set of static or dynamic rules $\mathcal{R}$. Again we would like to mention that we concentrate on the analysis
104 of the effects of applying different rules $\mathbf{R}$ and not on the very interesting but also wide field of
105 deriving the best rules $\mathcal{R}$ from $I$, see some discussion in Section 6. Nonetheless, we always motivate
106 the choice of the rules derived by $\mathbf{I}$.

107 **Rule Based Layers**  We now give a formal definition of rule based layers. Given some dataset
108 $(\mathbf{D}, \mathbf{L}, \mathbf{I})$ defined as before and the rule set $\mathcal{R}$ derived from $\mathbf{I}$, the task is to learn the weights $\Theta$ of
109 the neural network $\mathbf{f}$ to predict the labels of unseen examples drawn from an unknown distribution.
110 Our contribution concentrates on single layers and is fully compatible with other layers such as
111 linear layers, convolutional layers Hence, in the following we restrict to the $i$-th layer $f^i(-, \Theta^i, \mathbf{R}^i) :$
112 $\mathbb{R}^* \longrightarrow \mathbb{R}^*$ of a network $\mathbf{f}$. For simplicity, we assume $i = 1$ and omit the indices, i.e., we write
113 $f := f^i, \Theta := \Theta^i$ and $\mathbf{R} := \mathbf{R}^i$. The forward propagation step of the rule based layer $f$ which will be
114 a generalization of certain known layers as shown in Section 3 is as follows. Fix some input sample
115 $x \in \mathbf{D}$ with $x \in \mathbb{R}^n$. Then $f(-, \Theta, \mathbf{R}) : \mathbb{R}^n \longrightarrow \mathbb{R}^m$ for $n, m \in \mathbb{N}$ is given by

$$f(x, \Theta, \mathbf{R}) = \sigma(W_{\mathbf{R}_W(x)} \cdot x + b_{\mathbf{R}_b(x)}) \ . \tag{1}$$

116 Here $\sigma$ denotes an arbitrary activation function and $W_{\mathbf{R}_W(x)} \in \mathbb{R}^{m \times n}$ rsp. $b_{\mathbf{R}_b(x)} \in \mathbb{R}^m$ is some
117 weight matrix rsp. weight vector depending on the input vector $x$ and the rule $\mathbf{R}$. The set $\Theta :=$
118 $\{w_1, \ldots, w_N, b_1, \ldots, b_M\}$ consists of all possible learnable parameters of the layer. The parameters
119 $\{w_1, \ldots, w_N\}$ are possible entries of the weight matrix while $\{b_1, \ldots, b_M\}$ are possible entries of
120 the bias vector. The key point here is that the rule $\mathbf{R}$ determines the choices and the positions of
121 the weights from $\Theta$ in the weight matrix $W_{\mathbf{R}_W(x)}$ and the bias vector $b_{\mathbf{R}_b(x)}$ depending on the input

---

[1] Our code, results and the datasplits used can be found here.

122 sample $x$. More precisely, *not* all learnable parameters must be used in the weight matrix and the
123 bias vector for some input sample $x$. Note that for two samples $x, y \in \mathbf{D}$ of different dimensionality,
124 e.g., $x \in \mathbb{R}^n$ and $y \in \mathbb{R}^k$ with $n \neq k$ the weight matrices $W_{\mathbf{R}_W(x)}$ and $W_{\mathbf{R}_W(y)}$ also have different
125 dimensions and the learnable parameters can be in totally different positions in the weight matrix.
126 This is where the rules $\mathbf{R}$ and their associated rule functions, see (2) below, come into play.

127 Given the set of learnable parameters $\Theta := \{w_1, \ldots, w_N, b_1, \ldots, b_M\}$, for each input $x \in \mathbb{R}^n$ the
128 rule $\mathbf{R}$ induces the following two rule functions

$$\mathbf{R}_W(x) : [m] \times [n] \longrightarrow \{0\} \cup [N] \quad \text{and} \quad \mathbf{R}_b(x) : [m] \longrightarrow \{0\} \cup [M] \tag{2}$$

129 where $m \in \mathbb{N}$ is the output dimension of the layer that can also depend on $x$. In the following we
130 abbreviate $\mathbf{R}_W(x)(i, j)$ by $\mathbf{R}_W(x, i, j)$ and $\mathbf{R}_b(x)(i)$ by $\mathbf{R}_b(x, i)$. We note that for simplicity we
131 assume that the matrix and vector indices start at 1 and not at 0. Using the associated rule functions (2)
132 we can construct the weight matrix resp. bias vector by defining the entry $(i, j) \in \mathbb{R}^{m \times n}$ in the $i$-th
133 row and the $j$-th column of the weight matrix $W_{\mathbf{R}(x)} \in \mathbb{R}^{m \times n}$ via

$$W_{\mathbf{R}_W(x)}(i, j) := \begin{cases} 0 & \text{if } \mathbf{R}_W(x, i, j) = 0 \\ w_{\mathbf{R}_W(x, i, j)} & \text{o.w.} \end{cases} \tag{3}$$

134 and the entry at position $k$ in the bias vector $b_{\mathbf{R}_b(x)} \in \mathbb{R}^m$ by

$$b_{\mathbf{R}_b(x)}(k) := \begin{cases} 0 & \text{if } \mathbf{R}_b(x, k) = 0 \\ b_{\mathbf{R}_b(x, k)} & \text{o.w.} \end{cases}. \tag{4}$$

135 Summarizing, the *rule based layer* defined in (1) is a standard feed-forward layer with the difference
136 that the weights in the weight matrix and the bias vector are determined by a predifined rule $\mathbf{R}$.
137 In fact, weight matrix and bias vector depend on the input and can contain shared weights. More
138 precisely, the rule controls the connection between the $i$-th input and the $j$-th output feature in the
139 weight matrix. A rule $\mathbf{R}$ is called *static* if it is independent of the input $x \in \mathbf{D}$, i.e., $\mathbf{R}(x) \equiv \mathbf{R}(y)$
140 for all inputs $x, y \in \mathbb{R} \in \mathbf{D}$ otherwise it is called *dynamic*. We call a rule based layer as defined in (1)
141 *static* if it is based on a static rule $\mathbf{R}$ and *dynamic* otherwise. We will show in Section 3 that rule
142 based layers generalize known concepts of neural network layers for specific rules $\mathbf{R}$. In fact, we
143 show that fully connected layers and convolution layers are static rule based layers. Examples of
144 dynamic rule based layers are given later on in Section 4. The back-propagation of such a layer can
145 be done as usual enrolling the computation graph of the forward step and applying iteratively the
146 chain rule to all the computation steps. We will not go into the details of this computation as it is
147 similar to many other computations using backpropagation with shared weights. For the experiments
148 we use the automatic backpropagation tool of PyTorch [16] which fully meets our requirements.

149 **Assumptions and Examples**  Rule based learning relies on the following two main assumptions:
150 $A1)$ There is a connection between the additional information or expert knowledge $\mathbf{I}$ and the used
151 rule $\mathbf{R}$ and $A2)$ The distribution of weights given by the rule $\mathbf{R}$ in the weight matrix $W_{\mathbf{R}(x)}$ improves
152 the predictive performance or increases the interpretability of the neural network. As stated before
153 we concentrate on the second assumption and consider different distribution of weights in the weight
154 matrix given by different rules. In fact, we assume without further consideration that it is possible to
155 derive a meaningful ruleset $\mathbf{R}$ from the additional information or expert knowledge $\mathbf{I}$. For example if
156 the dataset consists of images we can derive the "informal" rule that neighboured pixels are more
157 important than pixels far away and in case of chemical data there exists, e.g., the ortho-para rule for
158 benzene rings that makes assumptions about the influence of atoms for specific positions regarding
159 the ring. This rule was already learned by a neural network in [28]. It is another very interesting task
160 which is beyond the scope of this work how to formalize these "informal" rules or to learn the "best"
161 formal rules from the additional information $\mathbf{I}$.

162 In the following sections we focus on the concept of rule based layers and therefore for simplicity
163 and space reasons only consider the rule function of weight matrices. The rule function associated
164 with the bias vector can be constructed similarly. For simplicity, we write $\mathbf{R}$ instead of $\mathbf{R}_W$.

## 3   Theoretical Aspects of Rule Based Layers

In this section we provide a theoretical analysis of rule based layers and show that they generalize fully connected and convolutional layers. More precisely, we define two *static* rules $\mathbf{R}_{\mathrm{FC}}$ and $\mathbf{R}_{\mathrm{CNN}}$ and show that the rule based layer as defined in (1) based on $\mathbf{R}_{\mathrm{FC}}$ is a fully connected layer and the rule based layer based on $\mathbf{R}_{\mathrm{CNN}}$ is a convolutional layer. All the proofs can be found in the Appendix A.

**Proposition 1** *Let $f : \mathbb{R}^n \longrightarrow \mathbb{R}^m$ with*

$$f(y, \Theta, \mathbf{R}_{\mathrm{FC}}) = \sigma(W_{\mathbf{R}_{\mathrm{FC}}(x)} \cdot y)$$

*be a rule based layer of a neural network as defined in* (1) *(without bias term) with learnable parameters $\Theta = \{w_1, \ldots, w_{n \cdot m}\}$ and $y = \mathbf{f}^i(x)$ is the result of the first $i - 1$ layers. Then for the rule function $\mathbf{R}_{\mathrm{FC}}(x) : [m] \times [n] \to [m \cdot n]$ defined for all inputs $x$ as follows*

$$\mathbf{R}_{\mathrm{FC}} \coloneqq \mathbf{R}_{\mathrm{FC}}(x)(i, j) \coloneqq (i - 1) \cdot n + j,$$

*the rule based layer $f$ is equivalent to a fully connected layer with activation function $\sigma$.*

Proposition 1 shows that rule based layers generalize fully connected layers of arbitrary size without bias vector and can be easily adapted to include the bias vector. Hence, this shows that rule based layers generalize arbitrary fully connected layers. Moreover, fully connected layers are static rule based layers as the rule $\mathbf{R}_{\mathrm{FC}}$ is static because it does not depend on the particular input $x$.

**Proposition 2** *Let $f : \mathbb{R}^{n \cdot m} \longrightarrow \mathbb{R}^{(n-N+1) \cdot (m-N+1)}$ with*

$$f(y, \Theta, \mathbf{R}_{\mathrm{CNN}}) = \sigma(W_{\mathbf{R}_{\mathrm{CNN}}(x)} \cdot y)$$

*be a rule based layer of a neural network as defined in* (1) *(without bias term) and $W^i = \{w_1, \ldots, w_{N^2}\}$ be the set of learnable parameters. Then for the rule function $\mathbf{R}_{\mathrm{CNN}} : [(n - N + 1) \cdot (m - N + 1)] \times [n \cdot m] \to [N^2]$ defined by*

$$\mathbf{R}_{\mathrm{CNN}} \coloneqq \mathbf{R}_{\mathrm{CNN}}(x)(i, j) \coloneqq \begin{cases} \tau(i, j) & \textit{if } 0 < \gamma(i, j) < N \cdot n \textit{ and} \\ & \quad 0 < j \ (\mathrm{mod}\ n) - j + \gamma(i, j) < N \\ 0 & \textit{o.w.} \end{cases}$$

$$\begin{aligned} \textit{with} \quad \tau(i, j) &= \gamma(i, j) - ((\gamma(i, j) - 1)//n) \cdot (n - N) \\ \textit{and} \quad \gamma(i, j) &= j - ((i - 1)//(n - N + 1)) \cdot n + (i - 1) \ (\mathrm{mod}\ (n - N + 1)) \end{aligned}$$

*the rule based layer $f$ is equivalent to a convolution layer with quadratic kernel of size $N$ ($N < n$, $N < m$) and a stride of one over a two-dimensional image of size $n \times m$ (without padding and bias vector) with activation function $\sigma$. The notation $a//b$ denotes the integer division of two integers $a$ and $b$.*

Proposition 2 shows that rule based layers generalize 2D-image convolution without padding and bias term. By adaption of the rule function it is possible to include the bias vector and padding. Moreover, the result can be generalized to higher dimensions kernels, non-quadratic kernels and arbitrary input and output channels. Hence, rule based layers also generalize arbitrary convolutional layers. Convolutional layers are static rule based layers as the rule $\mathbf{R}_{\mathrm{CNN}}$ is static because it is independent of the input. The following result is a direct implication from Propositions 1 and 2.

**Theorem 1** *Rule based layers generalize fully connected and convolutional feed-forward layers. Moreover, both layers are static rule based layers.*

We claim that also other types of feed-forward layers can be generalized by rule based layers using appropriate rule functions. Because of space limitations we would rather present a specific application of dynamic rule based layers on graphs.

## 4   Rule Based Learning on Graphs

One of the main advantages of rule based layers as introduced in this work is that they give rise to a dynamic neural network architecture that is freely configurable using different rules. In fact, the network is independent of the dimension and structure of the input samples. Hence, a natural application of our approach is graph classification. We would like to emphasize that graph classification is only one of many possible applications of rule based layers. Other possible applications are node classification, regression tasks, graph embeddings or completely different data-structures.

**Graph Preliminaries** By a graph we mean a pair $G = (V, E)$ with $V$ denoting the set of nodes of $G$ and $E \subseteq \{\{i, j\} \mid i, j \in V\}$ the set of edges. We assume that the graph is undirected and does not contain self-loops or parallel edges. In case that it is clear from the context we omit $G$ and only use $V$ and $E$. The distance between two nodes $i, j \in V$ in a graph, i.e., the length of the shortest path between $i$ and $j$, is denoted by $d(i, j)$. A labeled graph is a graph $G = (V, E)$ equipped with a function $l : V \to \mathcal{L}$ that assigns to each node a label from the set $\mathcal{L} \subseteq \mathbb{N}$. In this paper the input samples corresponding to a graph $(V, E)$ are always vectors of length equal to $|V|$. In particular, the input vectors can be interpreted as signals over the graph and each dimension of the input vector corresponds to the one-dimensional input signal of a graph node.

## 4.1 Graph Rules

The example on molecule graphs in Figure 2 and Appendix A.4 motivates the intuition behind different graph specific rules that can be used to define a graph neural network based on rule layers. The underlying general scheme to define a rule based layer on graphs is as follows: Let $G = (V, E)$ be a graph and $l : V \to \mathcal{L}$ a permutation equivariant labeling function of the nodes, i.e., for some permutation $\pi$ of $V$ it holds $l(\pi(V)) = \pi(l(V))$. Assuming that input and output dimension of the layer is equal to $|V|$ the rule functions $\mathbf{R}$ as defined in (2) map each pair of nodes $(i, j) \in V \times V$ to an integer which is the index of the learnable parameter in the set of all learnable parameters. The mapping is injective based on the labels $l(i), l(j)$ and an additionally defined shared property between the nodes $i$ and $j$. Examples for such shared properties can be the distance between $i$ and $j$, the type of the edge connecting $i$ and $j$ or the information, that $i$ and $j$ are in one circle. As an example $\mathbf{R}_{\text{Mol}}$ as defined in Appendix A.4 is induced by the permutation equivariant function $l$ that maps each node to its atom label and the shared property between two nodes is the type of the edge connecting the nodes or the absence of an edge. Besides $\mathbf{R}_{\text{Mol}}$ the simple rule that is based on the given node labels in this paper we focus on three different rule based layers for graphs.

**Proposition 3** *Let $\pi$ be some permutation of the nodes of $G = (V, E)$ and $x$ its corresponding input vector. If $\mathbf{R}$ permutation equivariant, i.e., $\mathbf{R}(\pi(x))(i, j) = \mathbf{R}(x)(\pi(i), \pi(j))$ then the rule based layer is also equivariant under node permutations, i.e., $f(\pi(x), \Theta, \mathbf{R}_{\text{Mol}}) = \pi(f(x, \Theta, \mathbf{R}_{\text{Mol}}))$.*

**Weisfeiler-Leman Rule** Recent research has shown that the Weisfeiler-Leman labeling is a powerful tool for graph classification [18, 14, 2, 22]. Thus, we propose to use Weisfeiler-Leman labels as one option to define the rule based layer for graph classification. The Weisfeiler-Leman algorithm assigns in the $k$-th iteration to each node of a graph a label based on the structure of its local $k$-hop neighborhood, see [18]. Let $l(v)$ be the result of the $k$-th iteration of the Weisfeiler-Leman algorithm for some node $v \in V$. Then the Weisfeiler-Leman Rule $\mathbf{R}_{WL_{k,d}}$ assigns to each node pair $(i, j)$ an integer or zero based on the Weisfeiler-Leman labels $l(i), l(j)$ and the distance between the nodes $i$ and $j$. The result is zero if the distance between $i$ and $j$ is not between 1 and $d$. Note that we are not restricted to look at consecutive distances from 1 to $d$. It is also possible to look at certain distances only if the expert knowledge suggests it. In fact, $(i, j)$ and $(k, l)$ are mapped to the same integer if and only if $l(i) = l(k), l(j) = l(l)$ and the distance between $i$ and $j$ is equal to the distance between $k$ and $l$. The layer defined by this rule is related to ordinary message passing but messages can pass between nodes of arbitrary distances. For computational reasons in the experiments we restrict the maximum number of different Weisfeiler-Leman labels considered by some bound $L$. We relabel the most frequent $l - 1$ labels to $1, \cdots, l - 1$ and set all other labels to $l$. The corresponding layer is denoted by $f_{WL_{k,d,L}}$.

**Pattern Counting Rule** Beyond labeling nodes via the Weisfeiler-Leman algorithm, it is a common approach to use subgraph isomorphism counting to distinguish graphs [3]. This is in fact necessary as the 1-Weisfeiler-Leman algorithm is not able to distinguish some types of graphs, for example circular skip link graphs [4] and strongly regular graphs [2, 3]. Thus, we propose the pattern counting rule and show in Section 5 that RuleGNNs based on this rule are able to perform well on synthetic benchmark datasets while message passing models based on the Weisfeiler-Leman algorithm fail. In general, subgraph isomorphis counting is a hard problem [5], but for the real-world and synthetic benchmark graph datasets that are usually considered, subgraphs of size $k \in \{3, 4, 5, 6\}$ can be enumerated in a preprocessing step in a reasonable time, see Table 5. Given a set of patterns, say $\mathcal{P}$, we compute all possible embeddings of these patterns in the graph dataset in a preprocessing step. Then for each pattern $P \in \mathcal{P}$ and each node $i \in V$ we count how often the node $i$ is part of an

embedding of $P$. Using those counts we define a labeling function $l : V \to \mathcal{L}$. Two nodes $i, j \in V$ are mapped to the same label if and only if their counts are equal for all patterns in $\mathcal{P}$. Patterns that are often used in practice are small cycles, cliques, stars, paths, etc. The Pattern Counting Rule $\mathbf{R}_{\mathcal{P}_d}$ assigns each node pair $(i, j)$ an integer or zero based on the values of $l(i), l(j)$ and the distance between $i$ and $j$. As for the Weisfeiler-Leman Rule we restrict the maximum number of different labels to some number $L$. The corresponding layer is denoted by $f_{\mathcal{P}_{d,L}}$.

**Summary Rule**  The summary rule $\mathbf{R}_{\mathrm{Out}}^N$ can be used as the output layer as its output is a fixed dimensional vector of size $N \in \mathbb{N}$ independent of the size of the input data and the output is invariant under node permutations. Again, let $l : V \to \mathcal{L}$ be a function that maps each node of a graph to some integer. Then the summary rule $\mathbf{R}_{\mathrm{Out}}^N$ assigns each pair $(n, i)$ with $i \in V$ and $n \in [N]$ an integer or zero based on $n$ and $l(i)$. In fact, for each element of $\mathcal{L}$ the rule defines $n$ different learnable parameters. The corresponding layer is denoted by $f_{\mathbf{R}_{\mathrm{Out}}^N}$.

All the above rules define dynamic rule based neural network layers because the weight matrix and bias terms defined by the rules depend on the input vectors $x$ corresponding to different graphs. Note that the layers defined by the above rules are permutation equivariant as the node labeling function $l$ used to define the rule is equivariant under node permutations. Thus, using the layers corresponding to the above defined rules we can build a graph classification architecture that by definition does not depend on the order of the nodes in the input graphs. Moreover, a layer is able to pass information between nodes of arbitrary distances in the graph. Thus, as shown in the experiments below, it is not necessary to use deep networks to achieve good performance on the real-world benchmark datasets.

## 4.2 Rule Graph Neural Networks (RuleGNNs)

The layers derived from the above rules are the building blocks of the RuleGNNs. Each RuleGNN is a concatenation of different rule based layers from Weisfeiler-Leman rules and pattern counting rules followed by a summary rule using arbitrary activation functions. To define the learnable parameters of the bias term we also use the summary rule. The input of the network is a signal $x \in \mathbb{R}^{|V|}$ corresponding to a graph $G = (V, E)$. We note that for simplicity we focus on one-dimensional signals but also multidimensional signals, i.e., $x \in \mathbb{R}^{|V| \times d}$ are possible. The output of the network is a vector of fixed size $N \in \mathbb{N}$ determined by the summary rule where $N$ is usually the number of classes of the graph classification task. The output can be also used as an intermediate vectorial representation of the graph or for regression tasks.

# 5 Experiments

We evaluate the performance of RuleGNNs on different real-world and synthetic benchmark graph dataset and compare the results to the state-of-the-art graph classification algorithms. For comparability and reproducibility of the results, also with future algorithms, we make use of the experimental setup from [7]. That means, for each graph dataset we perform a 10-fold cross validation, i.e., we use fixed splits of the dataset into 10 equally sized parts (the splits can be found in our repository), and use 9 of them for training, parameter tuning and validation. We then use the model that performs best on the validation set and report the performance on the previously unseen test set. We train the best model 3 times and average the results on each fold to decrease random effects. The standard deviation reported in the tables is computed over the results on the 10 folds.

**Data and Algorithm Selection**  A problem of several heavily used graph benchmark datasets like MUTAG or PTC [13] is that node and edge labels seems to be more important than the graph structure itself, i.e., there is no significant improvement over simple baselines [17]. Moreover, in case of MUTAG the performance of the model is highly dependent on the data split because of the small number of samples. Thus, in this work for benchmarking we choose DHFR, Mutagenicity, NCI1, NCI109, IMDB-BINARY and IMDB-MULTI from the TU Dortmund Benchmark Graphs repository [13] because the structure of the graphs seems to play an important role, i.e., the simple baselines presented in [17, 7] are significantly worse than the state-of-the-art graph classification algorithms. Additionally, we consider circular skip link graphs CSL [4] and constructed some new synthetic benchmark graph datasets called LongRings, EvenOddRings and Snowflakes [15] to show the advantages of RuleGNNs on more complex graph structures with given expert knowledge. For

| | NCI1 | NCI109 | Mutagenicity | DHFR | IMDB-B | IMDB-M |
|---|---|---|---|---|---|---|
| Baseline (NoG) [17] | $69.2 \pm 1.9$ | $68.4 \pm 2.2$ | $74.8 \pm 1.8$ | $71.8 \pm 5.3$ | $71.9 \pm 4.8$ | $47.7 \pm 4.0$ |
| WL-Kernel[18] | $\mathbf{85.2 \pm 2.3}$ | $\mathbf{85.0 \pm 1.7}$ | $\mathbf{83.8 \pm 2.4}$ | $83.5 \pm 5.1$ | $71.8 \pm 4.5$ | $51.9 \pm 5.6$ |
| DGCNN[27] | $76.4 \pm 1.7$ | $73.0 \pm 2.4$ | $77.0 \pm 2.0$ | $72.6 \pm 3.1$ | $69.2 \pm 3.0$ | $45.6 \pm 3.4$ |
| DGCNN (features) | $73.6 \pm 1.0$ | $72.5 \pm 1.5$ | $76.3 \pm 1.2$ | $76.1 \pm 3.4$ | $69.1 \pm 3.5$ | $45.8 \pm 2.9$ |
| GraphSage[8] | $76.0 \pm 1.8$ | $77.1 \pm 1.8$ | $79.8 \pm 1.1$ | $80.7 \pm 4.5$ | $68.8 \pm 4.5$ | $47.6 \pm 3.5$ |
| GraphSage (features) | $79.4 \pm 2.2$ | $78.6 \pm 1.6$ | $80.1 \pm 1.3$ | $82.4 \pm 3.9$ | $69.7 \pm 3.1$ | $46.6 \pm 4.8$ |
| GIN[26] | $80.0 \pm 1.4$ | $79.7 \pm 2.0$ | $81.9 \pm 1.4$ | $79.1 \pm 4.4$ | $71.2 \pm 3.9$ | $48.5 \pm 3.3$ |
| GIN (features) | $77.3 \pm 1.8$ | $77.7 \pm 2.0$ | $80.6 \pm 1.3$ | $81.8 \pm 5.1$ | $70.9 \pm 3.8$ | $48.3 \pm 2.7$ |
| GSN (paper) [3] | $83.5 \pm 2.3$ | - | - | - | $77.8 \pm 3.3$ | $54.3 \pm 3.3$ |
| CIN (paper) [1] | $83.6 \pm 1.4$ | $84.0 \pm 1.6$ | - | - | $75.6 \pm 3.7$ | $52.7 \pm 3.1$ |
| SIN (paper)[2] | $82.7 \pm 2.1$ | - | - | - | $75.6 \pm 3.2$ | $52.4 \pm 2.9$ |
| PIN (paper) [22] | $85.1 \pm 1.5$ | $84.0 \pm 1.5$ | - | - | $76.6 \pm 2.9$ | - |
| **RuleGNN** | $82.8 \pm 2.0$ | $83.2 \pm 2.1$ | $81.5 \pm 1.3$ | $\mathbf{84.3 \pm 3.2}$ | $\mathbf{75.4 \pm 3.3}$ | $\mathbf{52.0 \pm 4.3}$ |

Table 1: Test set performance of several state-of-the-art graph classification algorithms averaged over three different runs and 10 folds. The $\pm$ values report the standard deviation over the 10 folds. The overall best results are colored red and the best ones obtained for the fair comparison from [7] are in bold. The (features) variant of the algorithms uses the same information as the RuleGNN as input features additionally to node labels. The (paper) results are taken from the respective papers and might be obtained with different splits of the datasets.

more details on the datasets see Appendix A.5. For NCI1, IMDB-BINARY and IMDB-MULTI we use the same splits as in [7] and for CSL we use the splits as in [6] and a 5-fold cross validation. We evaluate the performance of the RuleGNNs on these datasets and compare the results to the baselines from [7] and [17] and the Weisfeiler-Leman subtree kernel (WL-Kernel) [18] which is one of the best performing graph classification algorithm besides the graph neural networks. For comparison with state-of-the-art graph classification algorithms we follow [7] and compare to DGCNN [27], GIN [26] and GraphSAGE [8]. Additionally, we compare to the results of some newer state-of-the-art graph classification algorithms [3, 1, 2, 22]. For the latter we use the results from the respective papers that might be obtained with different splits of the datasets.

**Experimental Settings and Resources** All experiments were conducted on a AMD Ryzen 9 7950X 16-Core Processor with 128 GB of RAM. For the competitors we use the implementations from [7]. For the real-world datasets we were not aware of expert-knowledge, hence we tested different rules and combinations of the layers defined in Section 4.1. More details on the tested hyperparameters can be found in Appendix A.7. We always use tanh for activation and the Adam optimizer [11] with a learning rate of $0.05$ (real-world datasets) resp. $0.1$ (synthetic datasets). For the real-world datasets the learning rate was decreased by a factor of $0.5$ after each 10 epochs. For the loss function we use the cross entropy loss. All models are trained for 50 (real-world) resp. 200 (synthetic) epochs and the batch size was set to 128. We stopped if the validation accuracy did not improve for 25 epochs.

**Real-World Datasets** The results on the real-world datasets (Table 1) show that RuleGNNs are able to outperform the state-of-the-art graph classification algorithms in the setting of [7] even if we add all the additional label information that RuleGNNs use to the input features of the graph neural networks (see the (features) results in Table 1). This shows that the structural encoding of the additional label information is crucial for the performance of the graph neural networks and not replacable by using more input features. Moreover, the results show that the Weisfeiler-Leman subtree kernel is the best performing graph classification algorithm on NC1, NCI109 and Mutagenicity. For IMDB-BINARY and IMDB-MULTI our approach performs worse than the state-of-the-art graph classification algorithms that are not evaluated within the same experimental setup.

**Synthetic Datasets** The results on the synthetic benchmark graph dataset show that RuleGNNs outperform the state-of-the-art graph classification algorithms if expert knowledge is available even in the case that mesage passing is enough to solve the task. In fact, CLS and Snowflakes are not solvable by the message passing model because they are not distinguishable by the 1-WL test. The results on LongRings show that long range dependencies can be easily captured by RuleGNNs and also dependencies between nodes of different distances as in case of the EvenOddRings dataset can be encoded by appropriate rules.

| | LongRings | EvenOddRings | EvenOddRingsCount | CSL | Snowflakes |
|---|---|---|---|---|---|
| Baseline (NoG) [17] | $30.17 \pm 3.2$ | $22.25 \pm 3.0$ | $47.9 \pm 3.9$ | $10.0 \pm 0.0$ | $27.3 \pm 5.3$ |
| WL-Kernel [18] | $100.0 \pm 0.0$ | $26.83 \pm 4.2$ | $47.8 \pm 4.3$ | $10.0 \pm 0.0$ | $27.9 \pm 4.1$ |
| DGCNN [27] | $29.9 \pm 2.6$ | $28.4 \pm 2.5$ | $59.1 \pm 5.2$ | $10.0 \pm 0.0$ | $26.0 \pm 3.3$ |
| GraphSAGE [8] | $29.8 \pm 2.8$ | $24.9 \pm 2.7$ | $51.3 \pm 1.9$ | $10.0 \pm 0.0$ | $25.0 \pm 1.8$ |
| GIN [26] | $32.0 \pm 3.1$ | $26.8 \pm 2.5$ | $51.0 \pm 3.7$ | $10.0 \pm 0.0$ | $24.5 \pm 2.2$ |
| **RuleGNN** | $\mathbf{99.0 \pm 3.3}$ | $\mathbf{90.2 \pm 7.2}$ | $\mathbf{100.0 \pm 0.0}$ | $\mathbf{100.0 \pm 0.0}$ | $\mathbf{97.9 \pm 3.2}$ |

Table 2: Test set performance of several state-of-the-art graph classification algorithms averaged over three different runs and 10 folds. The $\pm$ values report the standard deviation over the 10 folds. The best results and our algorithm are highlighted in bold.

**Interpretability of the Rule Based Layers**  Each learnable parameter of RuleGNNs can be interpreted in terms of the importance of a connection between two nodes in a graph with respect to their labels and their shared property (in our case the distance). In Figures 1 and 6 we see how the network has learned the importance of different connections between nodes for different distances and labels.

# 6  Related Work, Limitations and Concluding Remarks

Dynamic neural networks have been proven to be more efficient, have more representation power and better interpretability than static neural networks [9]. Our approach can be seen as a sample dependent dynamic neural network as for each input sample the network structure is adapted. In contrast to other sample dependent dynamic neural networks [20, 24], our approach changes the layer structure based on a predefined rule instead of the whole architecture. The rule based layers of RuleGNNs use the Weissfeiler-Leman labeling algorithm and subgraph isomorphism counting which are both recently used concepts in graph classification algorithms [18, 3, 2, 1]. The challenge for graph neural networks is the heterogenicity of the input data and the lack of a fixed order of the input data. [19] proposes a dynamic neural network for graph classification that uses node and edge labels and is similar to our approach. In fact, they also show that their approach generalizes CNNs. In contrast, they do not provide a general scheme to encode expert knowledge into the network. Moreover, their approach is not able to encode long range dependencies in the graph using only one layer. There exist graph neural networks that have learned the ortho-para rule for molecules [28]. While the additional information used in these algorithms is mostly hard-coded, we are able to integrate arbitrary rules.

**Limitations**  *Input Features:* So far we have only considered 1-dimensional input signals and node labels, i.e., our experimental results are restricted to graphs that have no multi-dimensional node features. Additionally, we have not considered edge features in our rules. In principle, multi-dimensional node features and edge labels can be handled by our approach with the cost of increased complexity. *Space:* For each graph we need to precompute the pairwise distances and store the positions of the weights in the weight-matrix. This is a disadvantage for large and dense graphs as we need to store a large number of positions. For dense graphs the number of positions can be quadratic in the number of nodes. *Structure:* To define a meaningful rule for a layer the input and output features need to be logically connected. Fortunately, this is the case for graphs but this fact can be a limitation for other structures. *Combinatorics:* If it is not possible to define a formal rule given some informal expert knowledge the number of possible rules that have to be tested can be very large. Thus, it is an interesting question if it is possible to automatically learn a rule that captures the expert knowledge in the best way. *Implementation:* As stated in [9] there is a "gap between theoretical & practical efficiency" regarding dynamic neural networks, i.e., common libraries such as PyTorch or TensorFlow are not optimized for dynamic neural networks.

**Concluding Remarks**  We have introduced a new type of neural network layer that dynamically arranges the learnable parameters in the weight matrices and bias vectors according to a formal rule. On the one hand our approach generalizes classical neural network components such as fully connected layers and convolutional layers. On the other hand we are able to apply rule based layers to the task of graph classification showing that expert knowledge can be integrated into the learning process. Moreover, our approach gives rise to a more interpretable neural network architecture as every learnable parameter is related to a specific connection between input and output features.

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

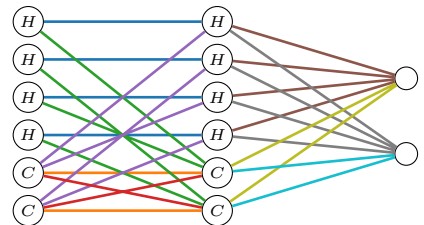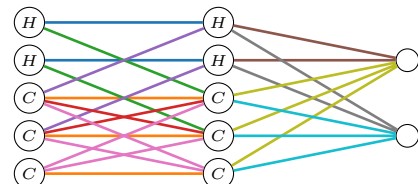

Figure 2: Information propagation in a simple two layer RuleGNN based on the molecule graphs of ethylene (left) and cyclopropenylidene (right) and the rules $\mathbf{R}_{\mathrm{Mol}}$ (5) and $\mathbf{R}_{\mathrm{Out}}$ (6). The input signal is propagated from left to right. The graph nodes represent the neurons of the neural network. Edges of the same color denote shared weights in a layer. For more details see Appendix A.4.

## A    Appendix / supplemental material

### A.1    Proof of Proposition 1

To show the equivalence between the two layers it suffices to show that their weight matrices coincide. In case of fully connected layers we have to show that the weight matrix $W_{\mathbf{R}_{\mathrm{FC}}(x)} \in \mathbb{R}^{m \times n}$ is filled with $n \cdot m$ distinct weights. This can be easily checked by computing $W_{\mathbf{R}_{\mathrm{FC}}(x)}$ using the definition of the weight distribution based on the rule function in (3).

### A.2    Proof of Proposition 2

Instead of the original two-dimensional image of size $n \times m$ we consider a reshaped vector $x \in \mathbb{R}^{n \cdot m}$ as our definition of rule based layers is restricted to simple vector matrix multiplication. The output vector of dimension $(n-N+1) \cdot (m-N+1)$ can then again be reshaped into a two-dimensional image of size $(n-N+1) \times (m-N+1)$. Unfortunately, the reshaping makes the rule function complicated as the indices of the reshaped vector have to be mapped to the indices of the two-dimensional image.

First note that convolution with a $N \times N$ kernel corresponds to matrix-vector multiplication of a doubly block circulant matrix that is a special case of a block Toeplitz matrix. Hence, to show the equivalence between the layers we have to compare the weight matrices and show that the entries in $W_{\mathbf{R}_{\mathrm{CNN}}(x)} \in \mathbb{R}^{(n-N+1) \cdot (m-N+1) \times n \cdot m}$ exactly matches the entries in the block Toeplitz matrix of the same dimension that corresponds to the convolution kernel. Comparing the definition of block Toeplitz matrices with the above given rule shows that the rule exactly returns the entries of the block Toeplitz matrix. Hence, the multiplication of $x$ with $W_{\mathbf{R}_{\mathrm{CNN}}(x)}$ is equivalent to multiplication of $x$ with the block Toeplitz matrix that is equivalent to the convolution of $x$ with a kernel of size $N \times N$.

### A.3    Proof of Proposition 3

The proof of Proposition 3 follows directly from the definitions of the rule based layers, see (1), and the rule functions, see (2). If the order of the nodes in the graph is permuted and rule function is permutation equivariant, then the node labels are permuted accordingly. Hence, the positions of the weights in the weight matrix and the bias term are permuted in the same way as the node labels. Thus, the result of $f$, i.e., the multiplication of permuted weight matrix with the permuted input signal, is the same as the permutation of the result of the multiplication of the original weight matrix with the original input signal.

### A.4    Example: RuleGNN for Molecule Graphs

Assume the task is to learn a property of a molecule based on its graph structure. In this example we present a RuleGNN that is a concatenation of two very simple rule based layers. The advantage of rule based layers and hence also RuleGNNs is that they encode the graph structure (in this example the structure of two molecules) directly into the neural network. Moreover, the input samples can be arbitrary molecule graphs and the output is a vector of fixed size $k$ that encodes the property of the molecule or some intermediate vectorial representation. In this example we consider the molecule graphs of ethylene and cyclopropenylidene given in Figure 3 together with their corresponding input

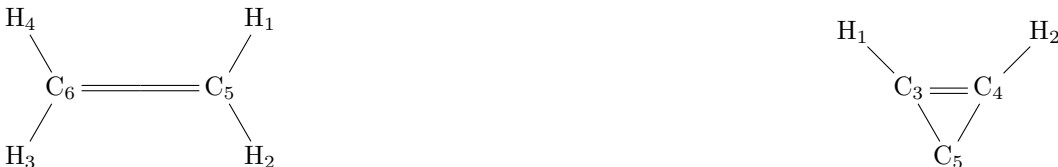

Figure 3: Molecule graphs of ethylene (left) and cyclopropenylidene (right). The indices denote the order of the nodes.

signals $x \in \mathbb{R}^6$ and $y \in \mathbb{R}^5$. The atoms of the molecules (hydrogen $H$ and carbon $C$) correspond to the nodes of a graph and the bond types (*single* and *double*) correspond to the edges. The atom labels and the atom bond types can be seen as additional information $\mathbf{I}$ that is known about the input samples. The graph nodes are indexed via integers in some arbitrary but fixed order and the atom corresponding to a graph node are given by the labeling function $l : V \to \{H, C\}$.

The RuleGNN consists of two rule based layers $f_1(, \Theta_1, \mathbf{R}_{\mathrm{Mol}})$ and $f(, \Theta_2, \mathbf{R}_{\mathrm{Out}})$ with learnable parameters $\Theta_1 = \{w_1, \ldots, w_6\}$ and $\Theta_2 = \{w'_1, \ldots, w'_{2 \cdot k}\}$ and the following rule functions $\mathbf{R}_{\mathrm{Mol}}$ and $\mathbf{R}_{\mathrm{Out}}$. For some graph $G = (V, E)$ and its corresponding input signal $z$ we define $\mathbf{R}_{\mathrm{Mol}}$ as follows:

$$
\mathbf{R}_{\mathrm{Mol}}(z): \quad [|V|] \times [|V|] \quad \longrightarrow \quad \{0\} \cup [6]
$$

$$
(i,j) \quad \mapsto \quad
\begin{cases}
1 & \text{if } i = j \text{ and } l(i) = H \\
2 & \text{if } i = j \text{ and } l(i) = C \\
3 & \text{if } (i,j) \text{ is an edge (-)}, l(i) = H, l(j) = C \\
4 & \text{if } (i,j) \text{ is an edge (-)}, l(i) = C, l(j) = H \\
5 & \text{if } (i,j) \text{ is an edge (-)}, l(i) = l(j) = C \\
6 & \text{if } (i,j) \text{ is an edge (=)}, l(i) = l(j) = C \\
0 & \text{o.w.}
\end{cases}
\tag{5}
$$

For some graph $G = (V, E)$ and its corresponding input signal $z$ we define $\mathbf{R}_{\mathrm{Out}}$ as follows:

$$
\mathbf{R}_{\mathrm{Out}}(z): \quad [|V|] \times [k] \quad \longrightarrow \quad \{0\} \cup [2 \cdot k]
$$

$$
(i,j) \quad \mapsto \quad
\begin{cases}
1 \cdot j & l(i) = H \\
2 \cdot j & l(i) = C \\
0 & \text{o.w.}
\end{cases}
\tag{6}
$$

Note that $\mathbf{R}_{\mathrm{Mol}}$ and $\mathbf{R}_{\mathrm{Out}}$ are not restricted to the two molecules from above but can be applied to arbitrary molecule graphs. Indeed, applying it to molecules with atom labels different from $H$ or $C$ makes the rules less powerful, i.e., it should be adapted to the type of molecules. Using the definition (3) of weight distribution defined by the rule function we can construct the weight matrices $W_{\mathbf{R}_{\mathrm{Mol}}(x)}, W_{\mathbf{R}_{\mathrm{Out}}(x)}$ for the ethylene graph and $W_{\mathbf{R}_{\mathrm{Mol}}(y)}, W_{\mathbf{R}_{\mathrm{Out}}(y)}$ for the cyclopropenylidene graph as follows:

$$
W_{\mathbf{R}_{\mathrm{Mol}}(x)} =
\begin{pmatrix}
w_1 & 0 & 0 & 0 & w_3 & 0 \\
0 & w_1 & 0 & 0 & w_3 & 0 \\
0 & 0 & w_1 & 0 & 0 & w_3 \\
0 & 0 & 0 & w_1 & 0 & w_3 \\
w_4 & w_4 & 0 & 0 & w_2 & w_5 \\
0 & 0 & w_4 & w_4 & w_5 & w_2
\end{pmatrix}
\qquad
W_{\mathbf{R}_{\mathrm{Out}}(x)} =
\begin{pmatrix}
w'_1 & w'_1 & w'_1 & w'_1 & w'_2 & w'_2 \\
\vdots & \vdots & \vdots & \vdots & \vdots & \vdots \\
w'_{2k-1} & w'_{2k-1} & w'_{2k-1} & w'_{2k-1} & w'_{2k} & w'_{2k}
\end{pmatrix}
$$

$$
W_{\mathbf{R}_{\mathrm{Mol}}(y)} =
\begin{pmatrix}
w_1 & 0 & w_3 & 0 & 0 \\
0 & w_1 & 0 & w_3 & 0 \\
w_4 & 0 & w_2 & w_6 & w_5 \\
0 & w_3 & w_6 & w_2 & w_5 \\
0 & 0 & w_5 & w_5 & w_2
\end{pmatrix}
\qquad
W_{\mathbf{R}_{\mathrm{Out}}(y)} =
\begin{pmatrix}
w'_1 & w'_1 & w'_2 & w'_2 & w'_2 \\
\vdots & \vdots & \vdots & \vdots & \vdots \\
w'_{2k-1} & w'_{2k-1} & w'_{2k} & w'_{2k} & w'_{2k}
\end{pmatrix}
$$

Combining the two rule based layers we obtain the RuleGNN and the forward propagation is given by $\sigma(W_{\mathbf{R}_{\mathrm{Out}}(x)} \cdot \sigma(W_{\mathbf{R}_{\mathrm{Mol}}(x)} \cdot x))$ for the ethylene graph and $\sigma(W_{\mathbf{R}_{\mathrm{Out}}(y)} \cdot \sigma(W_{\mathbf{R}_{\mathrm{Mol}}(y)} \cdot y))$ for the cyclopropenylidene graph.

| Dataset | #Graphs | #Nodes | | | #Edges | | | Diameter | | | #Node Labels | #Classes |
|---|---|---|---|---|---|---|---|---|---|---|---|---|
| | | max | avg | min | max | avg | min | max | avg | min | | |
| NCI1 | 4 110 | 111 | 29.9 | 3 | 119 | 32.3 | 2 | 45 | 11.5 | 0 | 37 | 2 |
| NCI109 | 4 127 | 111 | 29.7 | 4 | 119 | 32.1 | 3 | 61 | 11.3 | 0 | 38 | 2 |
| Mutagenicity | 4 337 | 417 | 30.3 | 4 | 112 | 30.8 | 3 | 41 | 6.3 | 0 | 14 | 2 |
| DHFR | 756 | 71 | 42.4 | 20 | 73 | 44.5 | 21 | 22 | 14.6 | 8 | 9 | 2 |
| IMDB-BINARY | 1 000 | 136 | 19.8 | 12 | 1249 | 96.5 | 26 | 2 | 1.9 | 1 | 1 | 2 |
| IMDB-MULTI | 1 500 | 89 | 13.0 | 7 | 1467 | 65.9 | 12 | 2 | 1.5 | 1 | 1 | 3 |

Table 3: Details on the real-world datasets used in the experiments. The datasets are from the TU Dortmund Graph Database [13].

Note that the forward propagation of the layer corresponding to the rule $\mathbf{R}_{\mathrm{Mol}}$ is kind of a multiplication with a weighted adjacency matrix of the graph where the weights of the adjacency matrix are given by the learnable parameters, see also Figure 2. In contrast to adjacency matrices the weight matrix is not necessary symmetric. The computation graph induced by the weight matrix exactly represent the graph structure while the edge weights are shared across the network using the rule, see Figure 2. Note that the above defined rule is very flexible as also edge labels (e.g., atomic bonds) can be taken into account by increasing the size of the weight set. Moreover, it is possible to include bigger neighbourhoods, i.e., all nodes reachable by $k$-hops. Of course using other information of the graph (e.g., substructures (such as circles or cliques), node degrees, connections not depicted by edges) more complicated rules can be defined.

## A.5 Dataset Details

In this section we provide additional details on the datasets used in the experiments. Table 3 shows an overview of the real-world datasets and Table 4 provides an overview of the synthetic datasets.

We consider the following synthetic datasets. The CSL dataset is from []. We constructed the other datasets to demonstrate the strength of our approach to encode expert knowledge into the neural network.

**LongRings** *LongRings* consists of 1200 cycles of 100 nodes each. Four nodes are labeled by $1, 2, 3, 4$ and all other nodes are labeled by 0. The distance between each pair of the four nodes is exactly 25 or 50. The label of the graph is 0 if 1 and 2 have distance 50, 1 if 1 and 3 have distance 50 and 2 if 1 and 4 have distance 50. There are 400 graphs for each class. The difficulty of the classification task is that information has to be propagated over a long distance. Regarding RuleGNNs this is very easy because if the expert knows that distance 50 is relevant we can define an appropriate rule.

**EvenOddRings** *EvenOddRings* consists of 1200 cycles of 16 nodes each. The nodes in each graph are labeled from 0 to 15. The graph label is based on the labels of the nodes that have distance 8 respectively 4 to the node with label 0. We denote them by $x$ resp. $y, z$. We distinct four cases: $x$ is even and $y + z$ is even, $x$ is even and $y + z$ is odd, $x$ is odd and $y + z$ is even, $x$ is odd and $y + z$ is odd. There are 300 graphs for each class, i.e., each of the four cases. The expert knowledge we use is that the information has to be collected from nodes of distance 8 and 4.

**EvenOddRingsCount** *EvenOddRingsCount* consists of the same graphs as EvenOddRings but the graph labels are different. For all nodes and their opposite node in the circle the sum of the labels is computed. If there are more even sums than odd sums the graph is labeled by 0 and by 1 otherwise. There are 600 graphs for each class. The expert knowledge we use is the information that only distance 8 is relevant.

**Snowflakes** *Snowflakes* is a dataset consisting of graphs proposed by [15] that are not distinguishable by the 1-WL test, see Figure 4 for an example. The dataset consists of circles of length 3 to 12 and at each circle node a graph from $M_0, M_1, M_2$ or $M_3$ is attached, see Figure 5 and [15] for the details. $M_0, M_1, M_2$ and $M_3$ are non-isomorphic graphs that are not distinguishable by the 1-WL test. One label in the circle is labeled by 1 and all other nodes are labeled by 0. The label of the graph is determined by the graph $M_0, M_1, M_2$ or $M_3$ that is attached to the circle node with label 1.

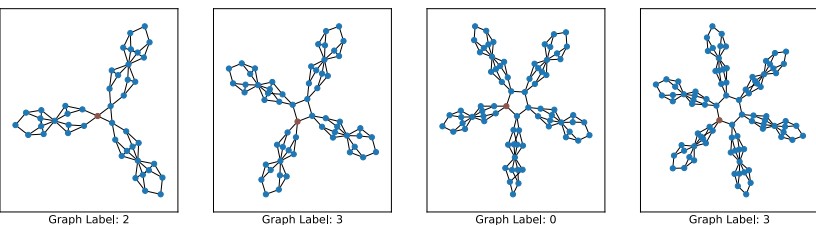

| Graph Label: 2 | Graph Label: 3 | Graph Label: 0 | Graph Label: 3 |

Figure 4: Example graphs from the *Snowflakes* dataset.

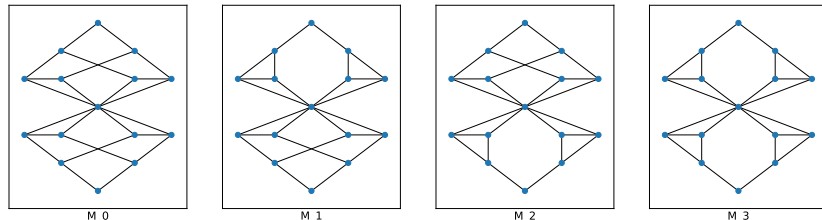

| M_0 | M_1 | M_2 | M_3 |

Figure 5: The graphs $M_0, M_1, M_2, M_3$ from [15] that are not distinguishable by the 1-WL test.

| Dataset | #Graphs | #Nodes | | | #Edges | | | Diameter | | | #Node Labels | #Classes |
|---|---|---|---|---|---|---|---|---|---|---|---|---|
| | | max | avg | min | max | avg | min | max | avg | min | | |
| LongRings | 1 200 | 100 | 100.0 | 100 | 100 | 100.0 | 100 | 50 | 50.0 | 50 | 5 | 3 |
| EvenOddRings | 1 200 | 16 | 16.0 | 16 | 16 | 16.0 | 16 | 8 | 8.0 | 8 | 16 | 4 |
| EvenOddRingsCount | 1 200 | 16 | 16.0 | 16 | 16 | 16.0 | 16 | 8 | 8.0 | 8 | 16 | 2 |
| CSL | 150 | 41 | 41.0 | 41 | 82 | 82.0 | 82 | 10 | 6.0 | 4 | 1 | 10 |
| Snowflakes | 1 000 | 180 | 112.5 | 45 | 300 | 187.5 | 75 | 18 | 15.5 | 13 | 2 | 4 |

Table 4: Details of the synthetic datasets used in the experiments. The CSL dataset is from [4].

## A.6 RuleGNNs: Runtimes

Table 5 shows more details of the RuleGNN model. In particular, we see that except for the DHFR dataset we need less than 12 epochs on average to reach the best result. This shows that our approach is very efficient and converges quickly. At the first glance the average time per epoch seems to be very high. This has two reasons. One is also mentioned in [9] that there is a gap between the theoretical and practical runtime of dynamic neural networks because the implementation in PyTorch is not optimized for dynamic neural networks. The other reason is that we parallelized the computation, i.e., we are able to run all the three runs and 10 folds in parallel on the same machine. Of course, this produces some overhead. As stated above the preprocessing times are not relevant for the experiments as they are only needed once. The third column shows the time needed to compute all the pairwise distances between the nodes of the graph. The fourth column shows the time needed to compute the node labels used for the best model. The most preprocessing time is needed for IMDB-BINARY and IMDB-MULTI because the graphs are much denser than the other datasets. For the synthetic datasets except for CSL we do not need any label preprocessing time as the original node labels are used.

## A.7 RuleGNNs: Architectures and Hyperparameters

Table 6 provides an overview of the different architectures used in the experiments that achieved the best results. One advantage of our approach is that messages can be passed over long distances. Hence, except for the EvenOddRings dataset we used only one layer and the output layer. In case of NCI1, NCI109, Mutagenicity it turns out that the best model uses the Weisfeiler-Leman rule with $k = 2$ iterations. We restricted the number of maximum labels considered to 500 which results in 250000 learnable parameters for the weight matrix and 500 for the bias vector. For the output layer we used the bound of 50000 learnable parameters which was larger than the number of different Weisfeiler-Leman labels in the second iteration. Interestingly, for NCI1 and NCI109 the best validation accuracy was achieved if considering node pairs with maximum distance 10. In case

| Dataset | Best Epoch | Avg. Epoch (s) | Preproc. Distances (s) | Preproc. Labels (s) | Num. Graphs |
|---|---|---|---|---|---|
| NCI1 | $7.3 \pm 5.3$ | $377.1 \pm 20.7$ | 2.0 | 11.9 | 4 110 |
| NCI109 | $5.4 \pm 2.9$ | $386.7 \pm 1.9$ | 2.4 | 13.2 | 4 127 |
| Mutagenicity | $9.1 \pm 4.1$ | $575.8 \pm 66.4$ | 2.2 | 15.2 | 4 337 |
| DHFR | $23.1 \pm 14.6$ | $44.4 \pm 9.0$ | 0.7 | 3.1 | 756 |
| IMDB-BINARY | $11.3 \pm 4.6$ | $24.3 \pm 0.9$ | 0.2 | 206.5 | 1 000 |
| IMDB-MULTI | $6.7 \pm 3.5$ | $19.6 \pm 1.3$ | 0.2 | 195.0 | 1 500 |
| LongRings | $194.2 \pm 15.1$ | $0.7 \pm 0.2$ | 6.6 | - | 1 200 |
| EvenOddRings | $176.1 \pm 15.2$ | $1.2 \pm 0.3$ | 0.2 | - | 1 200 |
| EvenOddRingsCount | $199.0 \pm 0.0$ | $0.5 \pm 0.1$ | 0.1 | - | 1 200 |
| CSL | $49.0 \pm 0.0$ | $1.6 \pm 0.0$ | 0.1 | 11.8 | 150 |
| Snowflakes | $191.7 \pm 18.9$ | $0.5 \pm 0.1$ | 7.1 | - | 1 000 |

Table 5: Runtimes and preprocessing times of the different datasets used in the experiments. All values are averaged over the best runs. The first column shows the best epoch (highest validation accuracy), the second column shows the average time per epoch, the third column shows the time needed to compute all the pairwise distances between the nodes of the graph, the fourth column shows the time needed to compute the node labels used for the best model and the last column shows the number of graphs in the dataset.

of Mutagenicity the best model uses only node pairs with distance 3 although we also considered the hyperparameter $d = 10$. We also tested different small patterns, e.g., simple cycles, but they did not improve the results. For DHFR this was different as the best model uses the pattern (simple cycles with length at most 10) for the output layer. We also tested the Weisfeiler-Leman rule in this case but the validation accuracy was lower. For IMDB-BINARY and IMDB-MULTI the best model uses the pattern (simple cycles with length at most 10, triangle, edge). Note that the embedding of one edge as a pattern is equivalent to the degree of the node. We also tested the Weisfeiler-Leman rule but the validation accuracy was lower. All in all we considered many different rules from type Weisfeiler-Leman and patterns but of course we did not test all possible rules. A full list of tested hyperparameters can be found here. As a next step it would be interesting to consider more rules, rules that come from expert knowledge or also deeper architectures with more rule based layers concatenated. Regarding the number of learnable parameters we would like to mention that the number is relatively high but lots of parameters are not used in the weight matrix. Hence, it might be possible to prune the set of learnable parameters by removing those that are not used or those that have a small absolute value.

For the synthetic datasets we use "expert knowledge" to define the rules. Hence we did not tested other rules than those in Table 6. For LongRings, EvenOddRings and EvenOddRingsCount we used the original node labels for the rule based layers. Moreover, instead considering learnable parameters for all node pairs of certain labels with distance smaller or equal to $d$ we considered only the node pairs with distance $d$ (denoted by "only: $d$"). In case of EvenOddRings we used two layers. The first layer that considers only node pairs with distance 8 collects all the necessary information of opposite nodes. The second layer that considers only node pairs with distance 4 collects the information of the nodes that are 4 hops away from the nodes with label 0, see also Figure 6. For CSL we used as patterns all simple cycles with length at most 10. For the Snowflakes dataset we used the patterns cycle of length 4 and 5 and collect the information of the nodes that have pairwise distance 3. In this way the RuleGNN is able to distinguish the graphs $M_0, M_1, M_2$ and $M_3$ that are not distinguishable by the 1-WL test. In the output layer we used the Weisfeiler-Leman rule with $k = 2$ iterations to collect the relevant information from nodes with different Weisfeiler-Leman labels.

## A.8   RuleGNNs: Interpretability

One advantage of our approach is that each weight can be interpreted, i.e., we can see the relevance of two nodes $i, j$ in a graph with labels $l(i), l(j)$ and distance $d(i, j)$. Figure 6 shows an example of the learned parameters for some synthetic dataset. Figure 1 shows an example of the relevance of the weights for a graph from the DHFR dataset using the weights of the best model. Considering Figure 6b we can see that in the first layer the RuleGNN passes the messages between opposite nodes as given by the rule. In the second layer it has learned to collect the information from the nodes that have distance 4 to the node with label 0 (dark blue node) all other connections of distance 4 have a smaller weight.

| Dataset | Rules | k | d | L | #Learnable Parameters per Layer |
|---|---|---|---|---|---|
| NCI1 | wl | 2 | 10 | 500 | 2 500 500 |
| | wl | 2 | - | 50000 | 4 220 |
| NCI109 | wl | 2 | 10 | 500 | 2 500 500 |
| | wl | 2 | - | 50000 | 4 336 |
| Mutagenicity | wl | 2 | 3 | 500 | 750 500 |
| | wl | 2 | - | 50000 | 4 972 |
| DHFR | wl | 2 | 6 | 500 | 1 382 880 |
| | pattern: (simple_cycles $\leq$ 10) | - | - | - | 112 |
| IMDB-BINARY | pattern: (triangle, edge) | - | 2 | - | 963 966 |
| | pattern: (induced_cycles $\leq$ 5) | - | - | - | 990 |
| IMDB-MULTI | pattern: (triangle, edge) | - | 2 | - | 551 775 |
| | pattern: (triangle, edge) | 10 | - | - | 1 578 |
| LongRings | labels | - | only: 25 | - | 30 |
| | labels | - | - | - | 18 |
| EvenOddRings | labels | - | only: 8 | - | 272 |
| | labels | - | only: 4 | - | 272 |
| | labels | - | - | - | 68 |
| EvenOddRingsCount | labels | - | only: 8 | - | 272 |
| | labels | - | - | - | 34 |
| CSL | pattern: (simple_cycles $\leq$ 10) | - | - | - | 8930 |
| | pattern: (simple_cycles $\leq$ 10) | - | - | - | 950 |
| Snowflakes | pattern: (cycle_4, cycle_5) | - | only: 3 | - | 90 |
| | wl | 2 | - | - | 20 |

Table 6: Overview over the hyperparameters of the best models.

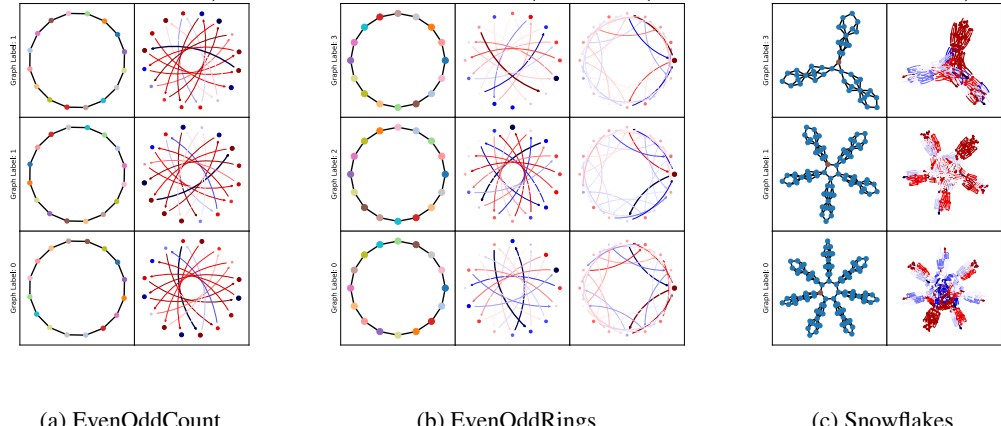

(a) EvenOddCount      (b) EvenOddRings      (c) Snowflakes

Figure 6: Visualization of the learned weights and biases for the RuleGNN on the EvenOddRingsCount (a), EvenOddRings (b) and Snowflakes (c) dataset. The first column shows the graphs and the colors of the nodes represent the different node labels. The other columns show the learned weights and biases of the RuleGNN for the respective rule based layer. The message passing weights are visualized by arrows (thicker for higher absolute values) and the biases are visualized by the size of the node (red for positive and blue for negative weights).

