# OpenReview forum: "Rule Based Learning with Dynamic (Graph) Neural Networks"
_NeurIPS.cc/2024/Conference — Submitted to NeurIPS 2024_

### Official Review · Reviewer_yEHT · 2024-07-07

**Soundness:** 3
**Presentation:** 3
**Contribution:** 3
**Rating:** 4
**Confidence:** 3

**Summary:**

The paper proposes a method for neural network-based learning to incorporate expert knowledge in the neural network architecture by building rules and utilizing them in "rule-based" layers of the learned neural networks. It introduces RuleGNNs as a concrete application of the proposed method and evaluates its performance against a few other SOTA methods. Empirical studies show competitive performance of RuleGNNs compared with other alternative methods.

**Strengths:**

- The idea of having dynamic rule-based layers in a neural network and especially for graph neural network learning is interesting. Although some existing methods in the literature including WL labeling could be considered doing the same, the proposed method builds on top of these building blocks and extends their ideas.
- Theoretical discussions in the paper and the assumptions behind them are clear.
- Experimental results cover an adequate set of alternative methods.

**Weaknesses:**

- The performance of RuleGNNs is expected to heavily rely on the quality of the rules generated from additional information or domain knowledge, however, the paper solely focuses on application of such rules without adequately discussing the challenges of building quality rules and feasibility of this fundamental step in the proposed method.
- Lack of clarity around how rules in RuleGNNs look like and how they can influence learning model parameters.
- Experimental results are not fully discussed. For example, WL-Kernel shows superior performance in three data sets and it would have been useful to provide more insights about what data set characteristics contribute to this.

**Questions:**

As stated under the weaknesses section, the main question from the reviewer is about where (i.e. for what type of data sets) one can expect RuleGNNs to outperform other alternatives methods, especially WL-Kernel which shows promising results and superior performance in a few tested data sets.

**Limitations:**

Authors have adequately addressed the limitations of their work by listing the following limitations:
- They have only considered 1 dimensional input signals and labels.
- They have not considered graphs with multi-dimensional node features.
- Edge features are not considered.
- Computation and storage limitation for large/dense graphs.
In addition, authors have clearly discussed structure, Combinatorics, and Implementation limitations of their work.

---

> ### Author Rebuttal · Authors · 2024-08-06
>
> ### Weaknesses:
>
> >W1: "The performance of RuleGNNs is expected to heavily rely on the quality of the rules generated from additional information or domain knowledge, however, the paper solely focuses on application of such rules without adequately discussing the challenges of building quality rules and feasibility of this fundamental step in the proposed method."
> >> The main focus of this work was to introduce a very general framework to incorporate expert knowledge into neural networks.
> > Thus, we did not consider the challenges of building quality rules in the paper.
> > We agree that the quality of the rules is crucial for the performance of RuleGNNs. We will extend the discussion on the challenges of building quality rules in a revised version of the paper.
>
> >W2:  "Lack of clarity around how rules in RuleGNNs look like and how they can influence learning model parameters."
> >> In 4.2 we describe the rules used for the experiments in Table 1 and Table 2, see also 4.2 "Each RuleGNN is a concatenation of different rule based layers from Weisfeiler-Leman rules and pattern counting rules followed by a summary rule using arbitrary activation functions."
> > The exact hyperparameters for the best performing RuleGNN can be found in the appendix of the paper in Table 6.
> > We did not report the results for different rules as our focus was on the comparison of RuleGNNs with other GNNs and not comparing the performances for different rules.
>
> >W3:  "Experimental results are not fully discussed. For example, WL-Kernel shows superior performance in three data sets and it would have been useful to provide more insights about what data set characteristics contribute to this."
> >> There is no easy answer to the question of why WL-Kernels are especially outperforming GNNs on these three data sets.
> > It seems that WL-labels generalize very well in this case. On other data sets where WL-labels are also able to distinguish the graphs in the graph dataset one can observe that WL-based methods overfit the data very quickly and hence do not generalize well.
> > A more detailed analysis of the data sets characteristics might be useful.
>
>
> ### Questions:
>
> >Q1: "As stated under the weaknesses section, the main question from the reviewer is about where (i.e. for what type of data sets) one can expect RuleGNNs to outperform other alternatives methods, especially WL-Kernel which shows promising results and superior performance in a few tested data sets."
> >> The simple answer to this question is that RuleGNNs outperform WL based methods if WL-labels are not sufficient to distinguish the graphs in the graph dataset (e.g., in case of the Snowflakes data set).
> > Moreover, RuleGNNs can capture long-range dependencies in the data or specific distances, see Table 2 that are not captured by alternative methods.
> > If graphs are distinguishable by WL-labels (e.g., in case of IMDB-BINARY, IMDB-MULTI) the answer why WL-Kernels are worse than GNNs is not that easy.
> > It is an interesting question how to choose the rules for the RuleGNNs given the data set characteristics.
> > Moreover, comparing different rules on the same data set and analyzing the results might be useful.

---

> > ### Comment · Reviewer_yEHT · 2024-08-11
> >
> > I'd like to thank the authors for their detailed explanations. After going through other reviewers' comments and authors' responses to them, I'm still unclear about how quality rules can be built and how they can affect the performance of the proposed method. Therefore, I'd like to change my score to "4: borderline reject".

---

### Official Review · Reviewer_kGau · 2024-07-09

**Soundness:** 2
**Presentation:** 1
**Contribution:** 2
**Rating:** 3
**Confidence:** 3

**Summary:**

This paper proposes a novel model architecture rule-based layer, which induces different parameters given different inputs. Theoretical analysis demonstrates how the proposed architecture reduces back to classical feed-forward layers, and empirical results on both synthetic and real-world data sets demonstrate that the proposed method can improve upon existing works.

**Strengths:**

The idea of rule-based layers and rule-based GNN is novel and interesting.

**Weaknesses:**

- The implementation in this work may need further elaboration to make the proposed method easier to understand.
- Empirical results may need further improvements to better support the proposed method.

**Questions:**

- The notations may require further explanation. Regarding the “additional information” $I$ introduced in Section 2, how do we assume the additional information can be used to derive a set of static or dynamic rules $R$? While the authors provide some rule examples in Appendix A.4, it would be better if the authors can provide some examples on the additional information as well.
- Also, how can we actually derive rules from the “additional information”? Do we need to manually design some rules or we need to apply some learning algorithms? While the authors claim it may not be the main focus of this paper, some clarification may still be needed to better understand the proposed method.
- I am also confused on how we obtain the parameters for the proposed rule-based GNN. Basically, compared with standard GNN models, I suppose the rules in $R$ enable us to use different parameters for different input $x$. Then do we simply learn these parameters by standard stochastic optimizers? The authors may need to explain more on that.
- While the proposed method seems to yield significant improvements in Table 2, the improvement in Table 1 for real-world data sets on the contrary seems quite marginal. As such, are there any other advantages of the proposed method compared with baseline methods in these tables? The authors may need to add more explanations on that.
- It seems also confusing why the authors do not explicitly mention the rules used for experiments in Table 1 and 2, which should be essential to understand how the proposed method performs well on these data sets.
- Moreover, no ablation studies or hyper-parameter sensitivity analysis is present in current submission. How do different rules (if they are manually set) or different learning algorithms affect the performance of proposed method? Such results and discussion are certainly needed to better understand the proposed method.

**Limitations:**

The authors discuss about possible limitations in the conclusion part, and no direct negative societal impact exists for this work from my perspective.

---

> ### Author Rebuttal · Authors · 2024-08-06
>
> ### Weaknesses:
>
> >W1: "The implementation in this work may need further elaboration to make the proposed method easier to understand."
> >
> >> As stated by the reviewer hyNs there are some minor issues that will be corrected in a revised version of the paper to improve the readability.
> >> Nevertheless, two of the reviewers found the paper easy to understand (reviewer yEHT: Strengths: "Theoretical discussions in the paper and the assumptions behind them are clear." and reviewer 9LgA: Strengths: "The presented theory is very general and simple").
>
> >W2:  "Empirical results may need further improvements to better support the proposed method."
> >
> >>As stated in the paper, the empirical results should only be seen as a proof of concept for the proposed framework.
> >>The main intention of the paper is to introduce a very general framework to incorporate expert knowledge into neural networks.
> >>The application to graphs shows that the framework can be applied in practice beating baseline GNN methods in terms of accuracy in the fair comparison.
> >>Moreover, the experiments on synthetic data show that the framework is more expressive than other GNN methods (see also Q4).
> >>We can provide more detailed results in a revised version of the paper.
>
>
> ### Questions:
>
> >Q1: Additional Information 𝐈
> >
> >>Additional Information can be any information about the data in any form, e.g., segmentation information for images or see section 2 "This can be for example knowledge about the graph structure, node or edge labels, importance of
> neighborhoods and many more."
> >>The question is how to derive rules from this information.
> >>In the paper, we do not provide a general framework to translate expert knowledge into rules (i.e. arbitrary functions) as this is a very broad field and depends on the specific application.
> >>Instead, we provide an example of how to use additional information in case of graphs to derive rules for graph classification (see Q2).
>
> >Q2: How to derive rules from the additional information?
> >
> >>We give no general method to derive rules from the additional information because there is no restriction on the form of additional information.
> >>Thus, rules need to be derived from the additional information in a way that is meaningful for the specific application.
> >>In the paper, we provide an example of how to use additional information in case of graphs to derive rules for graph classification.
>
> >Q3: How to obtain the parameters for the proposed rule-based GNN?
> >
> >>As described in the preliminaries section, we start from a set of learnable parameters that contain all the weights that can be used for weight matrices defined as W (layer-wise).
> >>The learning is via standard backpropagation and the parameters are updated using gradient descent as the forward pass is a simple matrix multiplication, see (1).
> >>The key point is that the rule gives the instruction on how to use the learnable parameters in the weight matrices (per input sample), see (2), (3) and (4) for the formal definitions.
>
> >Q4: Advantages over the baseline methods
> >
> >>In Table 1 we show that our method outperforms other GNN methods in the fair comparison in terms of accuracy.
> >>Besides the advantages in expressivity shown in Table 2, our framework gives rise to interpretable weights (see Figure 1, Figure 6 and paragraph "Interpretability of the Rule Based Layers" in 5) and allows to specify certain "important" distances in the graph (see the description in the paragraph "Weisfeiler-Leman Rule" in 4.1 and Figure 6).
> >>We could have made the advantages over the baseline methods more explicit in the paper but this was not the main focus of the work as we wanted to introduce a very general framework to incorporate expert knowledge into neural networks.
> >>The application to graphs was only one example of how to use the framework in practice.
>
>
> >Q5: Rules for the experiments in Table 1 and Table 2
> >
> >>In 4.1 we describe the rules used for the experiments in Table 1 and Table 2, see also 4.2 "Each RuleGNN is
> a concatenation of different rule based layers from Weisfeiler-Leman rules and pattern counting rules
> followed by a summary rule using arbitrary activation functions."
> >>The exact hyperparameters for the best performing RuleGNN can be found in the appendix of the paper in Table 6.
>
>
> >Q6: Ablation study on hyperparameters
> >
> >>As the application of our framework to graphs was only one example of how to use the framework in practice, we did not address the ablation study on hyperparameters in the paper.
> >>In fact, the results for different hyperparameters can be found in our repository.

---

> > ### Comment · Reviewer_kGau · 2024-08-11
> > **Acknowledging your rebuttal**
> >
> > I would like to first thank the authors for your detailed response. I appreciate that you have proposed a general framework for rule-based GNN. My key concern is that the proposed framework largely depends on the rule, which depends too much on appropriate expert knowledge and makes the effectiveness of proposed method questionable. Suppose we have enough expert knowledge, why should we choose the proposed framework instead of other ways to inject such knowledge (as is also mentioned by reviwer 9LgA)? Such dependency on expert knowledge is also partially reflected by results in Table 1 and Table 2, where the proposed framework achieves superior performance in Table 2 where rule can be directly induced, but not so in Table 1.
> >
> > Based on such concern and checking the reviews from other reviewers as well as your reply to them, I decided to keep my score and hope the authors can further improve the proposed framework based on current reviews.

---

### Official Review · Reviewer_hyNs · 2024-07-12

**Soundness:** 1
**Presentation:** 1
**Contribution:** 2
**Rating:** 3
**Confidence:** 4

**Summary:**

This paper introduces rule-based (dynamic) neural network layers. The basic idea is to have a common set of parameters, i.e., weights and biases, where, depending on a certain rule, only a subset of these parameters are used in the forward pass. They show that certain fully connected and convolutional layers can be regarded as a type of static rule-based neural network layer. In the remainder of the paper, the authors introduce three dynamic rules for graph classification tasks and perform experiments on synthetic and real-world datasets.

**Strengths:**

Overall, the concept of using rules based on expert knowledge to select different subsets of weights for various data samples or tasks seems useful and promising. This approach could offer significant benefits, such as training the same model on different tasks or on different datasets. Moreover, an approach which is able to learn on variably sized input data could be valuable on its own. The proposed rule-based layers for graph classification tasks outperform standard message-passing graph neural networks on synthetic and real-word datasets.

**Weaknesses:**

One of my primary concerns is that the main theoretical result of the paper, Theorem 1, is not proven. Specifically, while the authors show in Prop. 1 and Prop. 2 that fully connected layers _without bias_ and convolutional layers _without bias, padding, stride of one, and quadratic kernels_ can be expressed using their proposed (static) rule-based layer, the following paragraph leading to Theorem 1 claims this can be generalized to arbitrary convolutional layers. Although this might be straightforward to prove (and could be included in the appendix), the lack of a complete and formal proof severely undermines the soundness of the submission. If Theorem 1 is intended as a summary of Proposition 1 and Proposition 2, I suggest making this explicit by clearly stating the specific types of FC and CNN layers and renaming Theorem 1 to Corollary 1, or merging Prop. 1 and Prop. 2 into Theorem 1. Moreover, while the paper introduces some mathematical framework and formalizes existing concepts within this framework, it lacks proofs demonstrating what this framework can achieve and fails to establish connections to existing work. Given the lack of substantial theory, I think a more thorough empirical investigation could strengthen the submission. Comparisons with more expressive architectures are missing (e.g., in Table 1 there are no results reported for more expressive architectures for almost half of the datasets; for the synthetic datasets no comparison is done with more expressive architectures), making it difficult to appreciate the practical advantages of using the rule-based layers in practice. The practical relevance is limited further by the fact that the rule-based layer can only process one-dimensional features, and the higher space complexity for dense graphs.

Regarding clarity, there is considerable room for improvement. The concept of how a rule-based layer works was not fully clear to me until page 4. If my understanding is correct, we have a matrix $\mathcal{W}$ that contains all possible weights (and similarly a bias vector $\mathcal{B}$ with all possible biases). A rule restricts $\mathcal{W}$ to a subset of weights; applying rule *R* means setting some entries in $\mathcal{W}$ to zero. If my understanding is incorrect, this indicates that the writing lacks some clarity. I suggest shortening the introduction and preliminaries, which are at times verbose, and including a briefer example from Appendix A.4 earlier in the paper, or providing a clearer definition sooner. Additionally, the notation for the rule-based layer presentation is somewhat convoluted. The readability of the paper is also hindered by the inconsistent use of formal definitions and natural language. While both approaches can be fine (as long as they are precise), there is a noticeable mismatch between the rigor in the preliminaries and, for example, Section 4. Many aspects of the paper are thus unclear; please refer to the *Questions* and *Minor Remarks* for specific examples.

Overall, I think this paper presents promising ideas in a preliminary manner. As also stated by the authors, the dynamic rule-based layer seems to be reasonable for graphs, but is more difficult to devise for other structures. One approach could be to revise the paper from a graph learning perspective, and, if the authors have novel results which hold for general structures, present these results in a follow-up paper. Another exciting direction could be to use rules to create flexible machine learning models for different tasks and input data.

*Minor remarks*:

* line 33: each new information -> each new piece of information
* line 34: the essence -> a bit vague, what is the essence of dynamic NNs?
* Fig. 1 is too small and difficult to parse in general; there is also and typo in the last sentence
* line 75: dot missing after end of sentence
* line 95: concatentation -> should this be "composition"?
* line 111: dot missing after end of sentence
* Somewhat inflationary use of "respectively"
* line 123, 140: I would strongly advise to not use $y$ here for $x, y \in D$, as $y$ is already used to denote labels earlier
* It would be helpful to refer to equations as eq. (1) (instead of just (1))
* Could it simplify presentation if you define $\Theta$ as tuple $(\mathcal{W}, \mathcal{B})$?
* Last sentence of Prop. 1 is difficult to read
* Why do we call the learnable parameters $\Theta$ in Prop. 1 and $W^i$ in Prop. 2?
* line 190: higher dimensions -> higher dimensional
* line 202: network -> network architecture
* lines 206-214: I suggest to consider moving this to the preliminaries
* line 221: either rule function (singular) or rule functions R_W, R_B
* line 225: circle -> cycle
* Prop. 3: "its" -> not clear what it refers to
* line 231: If R permutation-equivariant -> language sounds off, maybe "For permutation-equivariant R" or "If R is permutation-equivariant"
* line 255: typo in isomorphism
* Pattern counting rule: $d$ is never defined
* line 347: missing space

**Questions:**

1. Is it possible to encode multiple rules into one layer? E.g., if a rule consists of a conjunction, are there beneftis/downsides to encoding the conjuncts in different layers or all in one?
2. Could you give more examples of how to exploit the fact that dynamic layers allow for arbitrarily sized input (beyond graphs)?
3. Could you elaborate on the precise meaning of "injective based on the labels and an additionally defined shared property"?
4. Does the presented approach allow to combine different data types/modalities and train them in the same network (towards, e.g., in-context learning)?
5. Did you experiment on strongly regular graphs? The ability to count cycles is helpful for that graph class and it could give some information about the expressivity of the rule-based graph layers.
6. Line 258: "We compute all possible embeddings". What does embedding refer to here?
7. Could you put your work into context with other modular (dynamic) neural network approaches (e.g., https://arxiv.org/pdf/1910.04915 or https://arxiv.org/pdf/2010.02066).
8. What happens if we want to classify a new sample, but we do not know what rule to use. How can we use the proposed approach in this scenario?

**Limitations:**

One of the main limitations, as the authors point out themselves, is that their proposed rule-based layer can only process one-dimensional node features, and no edge features, which impacts the practical value of their method. For more limitations, please refer to *Weaknesses* and *Questions*.

---

> ### Author Rebuttal · Authors · 2024-08-06
>
> ### Weaknesses:
> >W1: Theorem 1
> >>Indeed, Theorem 1 in this generality is not proven in the appendix, but it is straightforward to extend Proposition 2 to the mentioned cases.
> > As suggested, in a revised version of the paper, we will provide a full proof of Theorem 1 or restrict the statement of Theorem 1 to the assumptions made in Proposition 2 which is proven in the appendix.
>
> >W2:  Establish connections to existing work
> >> We agree that the paper could benefit from a more detailed discussion of the connections to existing work. We will extend the related work section in the revised version of the paper.
>
> >W3:  Comparisons with more expressive architectures
> >> We agree that the paper could benefit from a more detailed comparison with more expressive architectures.
> > We would like to point out that the goal of the paper was to provide a very general framework that can be used to incorporate expert knowledge in a flexible way.
> > Thus, the main focus of the paper was not a very detailed comparison with more expressive architectures for the example of graph classification.
>
> >W4:  One-dimensional features, and the higher space complexity
> >>Higher space complexity for dense graphs is indeed not a disadvantage of our framework compared to existing GNNs but rather a general limitation because of the high number of edges.
> > Indeed, using our framework is an advantage as we can choose a rule that give rise to a sparse weight matrix.
> > We do not have made this point clear enough in the paper that the limitation is only in the preprocessing of the data and not in the framework itself.
> > Regarding the one-dimensional features see the comment on L1.
>
> >W5: Clarity
> >>As stated in paragraph "Rule Based Layers" in section 2 we have a set of learnable parameters θ which contain all the weights that can be used for weight matrices defined as W (layer-wise).
> > Informally the rule is an instruction on how to use the learnable parameters θ in the weight matrices W (positions, weight sharing), see (2), (3) and (4) for the formal definitions.
> > That means we do not set entries to zero but dynamically build weight matrices for each input data based on Θ and the rule.
>
> >W6: Minor Remarks
> >>We will correct the typos and improve the readability of the paper in a revised version.
>
> ### Questions:
> >Q1: Multiple rules/conjuctions in one layer
> >>It is possible to encode multiple rules into one layer.
> >Our framework as defined in the paper is based on simple vector matrix multiplication.
> > Hence, to define multiple rules in one layer using vector matrix multiplication we need some kind of reshaping of the input features.
> >Another option is to change the definition towards a channel based approach like in convolutional neural networks.
> >Then each channel corresponds to a different rule.
> >It is an interesting idea for future work to compare conjuncts in different layers or all in one.
>
> >Q2: Arbitrarily sized input
> >>Example beyond graphs:
> >Assume that the data consists of arbitrarily sized images and for each image some pixels have additional labels (e.g. the center, the corners, segmented regions, etc.).
> > Our approach allows to map connections between labeled pixels to certain weights.
> > E.g. specific weights between pixels that are part of the same or different segmented regions which might help to interpret the relations between labeled pixels.
> >As the size of the weight matrix (filled with the fixed weights from the weight set) is dynamic and depends on the size of the input, we can use the dynamic layers to process images of arbitrary size.
>
> >Q3: Meaning of "injective based on the labels and an additionally defined shared property"
> >>The rule function is a mapping from V x V to some integer wich should be interpreted as the index of a weight in the weight set.
> > Given two node pairs (u,v) and (u',v') the rule function is injective, i.e., if l(u) ≠ l(u') or l(v) ≠ l(v') or the shared property is not the same, i.e., d(u,v) ≠ d(u',v') then the images (i.e. the corresponding weights) of the two node pairs under the rule function should be different.
>
> >Q4: Strongly regular graphs
> >>We did not experiment on strongly regular graphs but could add this case to the experiments in a revised version of the paper.
> > In fact, due to similar structural properties compared to CSL, we expect similar results as for CSL.
>
> >Q5: Meaning of embedding
> >>By embedding we mean one subgraph isomorphism of a pattern graph into a graph of the graph dataset.
>
> >Q6: Relations to other modular (dynamic) neural network approaches
> >>We have not considered the mentioned approaches but can put our work into context with other modular (dynamic) neural network approaches in a revised version of the paper.
>
> >Q7: Classifying new samples
> >>The rules are fixed in advance, see the definition of rule based layers in (1). To classify a new sample, we preprocess the data with the fixed rules, i.e., compute the weight matrices and bias vectors for the rule based layers for the new sample and then apply the forward pass of the neural network.
> > In this paper we do not have considered the problem of how to find the best rules for a completely new dataset because this is a very broad field and depends on the specific application.
>
> ### Limitations:
> >Indeed, we mentioned that our rule-based layer can only process one-dimensional node features and no edge features but this is not a limitation of the framework but rather a limitation of the specific rules we used in the paper.
> >As most of the graphs used in the experiments have only one-dimensional node features and no edge features, we did not consider more complex rules.
> >Due to the generality of the approach it is possible to define rules that encode edge features or even more complex features, e.g. counting edge features over paths, pass only messages between nodes that lie in cycles, pass only messages from a node not lying in a cycle to a node lying in a cycle, etc.

---

> > ### Comment · Reviewer_hyNs · 2024-08-09
> >
> > I thank the authors for carefully reading all reviews and answering the questions in detail. I read the authors' rebuttal and also the other reviews and would like to **maintain my current score**. In my opinion, the concerns raised by other reviewers and me (such as **lack of rigorous theoretical analysis of the introduced framework, no comparisons with more expressive architectures** as well as **practical limitations**) cannot be sufficiently addressed within the rebuttal period. However, I hope the authors can benefit from the reviews and have a stronger submission for a different venue in the future.
> >
> > Two additional comments on the authors' rebuttal:
> >
> > * Answer to Q2: I think this idea is very interesting and I encourage the authors to extend on this in a revised version of the paper to showcase that their proposed method works well beyond the graph domain.
> > * Answer to Q4: I encourage the authors to test on strongly regular graphs. Despite similar structural properties, these datasets differ in terms of expressivity; 3-WL can distinguish all pairs of non-isomorphic graphs in CSL, whereas SR graphs are 3-WL-equivalent.
> >
> > Overall, I still believe that focusing on just the graph domain could be a promising direction by itself as well (e.g., experiments on non-synthetic long-range interaction datasets and comparisons with more expressive architectures).

---

### Official Review · Reviewer_9LgA · 2024-07-15

**Soundness:** 3
**Presentation:** 1
**Contribution:** 1
**Rating:** 3
**Confidence:** 3

**Summary:**

The authors develop a broad framework for adding expert knowledge to Neural Networks. They formalize this by extending the learnable parameterized functions with an additional parameter consisting of the set of formal rules. In general, these rules maybe learnable as well. However, the authors focuses on these rules being given in the form of expert-knowledge. The authors then introduce the set of rule based layers. And shows that fully connected NN layers and CNN layers are special cases of the rule based layers. They introduce three rule based layers for graphs: Weisfeiler-Leman Layer, Pattern-Counting layer and Aggregation layer. The author shows that there exists a GNN with rule based layers that can distinguish any two isomorphic graphs. Finally the author introduces some examples of rule based layers for specific molecule graphs. And presents experimental results on synthetic and real-world data.

**Strengths:**

-- The idea of adding expert knowledge to NNs and GNNs specifically is quite interesting and widely investigated.

-- The presented theory is very general and simple

**Weaknesses:**

-- The author has used the notion of rules rather broadly. There is no formal language (logic or matrix language) for the rules. They are just arbitrary functions. This basically means that any existing NN model, in one way or another, can be seen as a special case of Rule based NN. In my understanding, this makes the introduced framework a rather simple formalization of how expert knowledge maybe added to NNs. However, this formalization is so loose, that it does not really admit any meaningful analysis or provide any meaningful guidance to the user for adding knowledge.

-- None of the examples presented by the author are beyond what would be anyway possible by adding some simple graph features to the node features. This could be an interesting direction to investigate. But just formally stating that this is possible is not very interesting.

**Questions:**

How does your framework help a practitioner, in a meaningful way, more than just augmenting feature vectors in GNNs with expert knowledge?
How does your framework enable any new theoretical analysis of GNNs?

**Limitations:**

The authors have indeed touched upon most of the points I mention as weaknesses.
However, as mentioned earlier, the proposed framework is very broad and does not provide a meaningful way to proceed.

---

> ### Author Rebuttal · Authors · 2024-08-06
>
> ### Weaknesses:
> > W1: "The author has used the notion of rules rather broadly. There is no formal language (logic or matrix language) for the rules. They are just arbitrary functions.
> > This basically means that any existing NN model, in one way or another, can be seen as a special case of Rule based NN."
> >>In fact, the goal of the paper was to provide a very general framework that can be used to incorporate expert knowledge in a flexible way. Indeed, the fact that "The presented theory is very general and simple" was also mentioned as a strength of the paper is somewhat contradictory to the above statement.
> > Yes, we defined rules as arbitrary functions (and not as formal language) to show how general the framework is and which seems at first sight too "loose". Because of this "looseness", on the one side we show that our very general framework allows to generalize existing NN models. (for us a strength)
> > On the other side we show an explicit application of the framework to graphs showing that there is not only a theoretical possibility but also a practical application. (for us a strength)
>
> >W2: "In my understanding, this makes the introduced framework a rather simple formalization of how expert knowledge maybe added to NNs."
> >>Again, we are not sure why the above statement is a weakness of the paper as our goal was to present a general framework/formalization to incorporate expert knowledge into neural networks.
> > You mention that "The idea of adding expert knowledge to NNs and GNNs specifically is quite interesting and widely investigated." is a strength of the paper and also that our framework is a "simple formalization of how expert knowledge maybe added to NNs".
> > Thus, taking into account the above statements, we do not see why the simplicity of the formalization is a weakness of the paper.
>
> >W3: "However, this formalization is so loose, that it does not really admit any meaningful analysis or provide any meaningful guidance to the user for adding knowledge."
> >>Indeed, the formalization of rules as arbitrary functions is very general which was intended to be so (see W1).
> > We do not provide a framework to translate expert knowledge into rules (i.e. arbitrary functions) as this is a very broad field and depends on the specific application.
> > Nevertheless, we do not agree that "it does not really admit any meaningful analysis or provide any meaningful guidance to the user for adding knowledge" as we provide an example of how to use the framework for graph classification.
> > Using different hyperparameters (rules) we are able to find rules that performs better than other rules, i.e., we are able to find rules that are meaningful for the specific application (see paragraph "Real-World Datasets" in 5).
> > Moreover, we are able to visualize and possibly reuse the learned weights for the rules (see Figure 1, Figure 6 and paragraph "Interpretability of the Rule Based Layers" in 5).
>
>
>
> >W4: "None of the examples presented by the author are beyond what would be anyway possible by adding some simple graph features to the node features. This could be an interesting direction to investigate. But just formally stating that this is possible is not very interesting."
> >>Indeed, adding some simple graph features to the node features is another way to incorporate expert knowledge into graph neural networks. This has several disadvantages compared to our approach that are already mentioned in the paper:
> >>1. We are losing the association between the weights and the input/output features, i.e., the weights are not interpretable anymore (see Figure 1, Figure 6 and paragraph "Interpretability of the Rule Based Layers" in 5).
> >>2. Our experiments show that it is not the same to add some simple graph features to the node features as we compared our approach to GNNs where we have exactly added the same information (features) to the GNNs as node features (see paragraph "Real-World Datasets" in 5 and the (features) results in Table 1).
> >>3. Our framework allows more than adding simple graph features to the node features as we only need one layer to pass the messages over arbitrary distances in the graph or we can specify certain "important" distances (see the description in the paragraph "Weisfeiler-Leman Rule" in 4.1 and Figure 6).
>
>
>
> ### Questions:
>
> >Q1: "How does your framework help a practitioner, in a meaningful way, more than just augmenting feature vectors in GNNs with expert knowledge? How does your framework enable any new theoretical analysis of GNNs?"
> >>Our framework is not intended to directly be used by a practitioner but rather by researchers to incorporate expert knowledge into neural networks in a very general and dynamic way.
> > It is not the goal of this work to translate expert knowledge into rules (i.e. arbitrary functions) in general, as this is a very broad field and depends on the specific application.
> > Thus, we introduce the example of graph classification to show some possibilities of how to use the framework in practice.
> > Our framework enables new theoretical analysis of GNNs in the way that it connects GNNs to a very general framework that can be applied to almost arbitrary domains.
>
>
> ### Limitations:
>
> >L1: "The authors have indeed touched upon most of the points I mention as weaknesses. However, as mentioned earlier, the proposed framework is very broad and does not provide a meaningful way to proceed."
> >>Regarding the first part of L1, we are not sure why the review has mentioned our limitations as weaknesses saying that "The authors have indeed touched upon most of the points I mention as weaknesses."
> > In fact, the review does not explain why the limitations we are considering are the reason for rejecting the paper.  Regarding the second part of L1, we do not agree that the proposed framework does not provide a meaningful way to proceed as we provide an example of how to use the framework for graph classification (see the comments on W3, W4 and Q1).

---

> > ### Comment · Reviewer_9LgA · 2024-08-09
> > **Thanks for the rebuttal**
> >
> > I thank the authors for the detailed response.
> > My fundamental concern regarding the paper remains that the paper does not really present a meaningful, designable or interpretable notion of rules.
> > For instance Figure 1 and Figure 6 (as pointed in the rebuttal), in my understanding, do not aid interpretability. Furthermore, Example in Figure 6 has a rather simple underlying rule, and the paper's method does not make this easily accessible to the user.
> >
> > Finally, I maybe wrong about comparing paper's method to simply adding node features. But I really have difficulty in understanding how this method helps in adding expert knowledge besides cases where one can easily spell out small amounts of expert knowledge already in some other way or form (eg. additional node features, additional subgraph count or distance oracles with learnable weights). This is also reflected in the paper's exceptional performance in Table 2, where all relevant knowledge is known before hand, and moreover it is known how to express it easily. In real-world cases, as indicated in some experiments in the appendix A.7, it is not always easy to find meaningful rules that lead to improvements. Furthermore, you also mention that the number of learnable parameters are high, but all of them are not used. This definitely hurts interpretability and I am not sure how easy or meaningful is deciding the threshold for weights with small absolute value (as mentioned in the appendix). As mentioned in the rebuttal:
> >
> > - The paper is not aimed at practitioners.
> > - The paper does not provide a general method to translate existing expert knowledge into rules that can be incorporated into NNs.
> >
> > Finally, in my understanding, the paper's methods cannot extract meaningful expert knowledge beyond what is possible by normal data analysis/visualization techniques.
> >
> > Regarding the ambivalence in my review, it stems from a general appreciation of the idea of adding expert knowledge to NNs and rule learning in general. However, the paper does not deliver an impactful step in this direction.

---

### Author Rebuttal · Authors · 2024-08-06

First of all, we would like to thank the reviewers for their valuable feedback and comments.
Regarding the reviewer specific comments, we have addressed point by point the reviewer's comments and answered the questions raised in the reviews in the "Rebuttal by Authors".
In the following, we will summarize and discuss the main points raised in the reviews regarding the strengths and weaknesses of the paper.

### Strengths:

The main strengths raised by the reviewers are the following:
1. Idea and concept of adding expert knowledge to NNs and GNNs using rule based layers:
    - "The idea of adding expert knowledge to NNs and GNNs specifically is quite interesting and widely investigated." (9LgA)
    - "seems useful and promising" (hyNs)
    - "The idea of rule-based layers and rule-based GNN is novel and interesting." (kGau)
    - "The idea of having dynamic rule-based layers in a neural network and especially for graph neural network learning is interesting." (yEHT)
2. Presented theory:
    - "The presented theory is very general and simple" (9LgA)
    - "Theoretical discussions in the paper and the assumptions behind them are clear" (yEHT)
    - " This approach could offer significant benefits, such as training the same model on different tasks or on different datasets." (hyNs)
3. Experiments:
    - "The proposed rule-based layers for graph classification tasks outperform standard message-passing graph neural networks on synthetic and real-word datasets." (hyNs)
    - "Experimental results cover an adequate set of alternative methods" (yEHT)


### Weaknesses:

We collected the main weaknesses raised by the reviewers and discuss them below:
1. Notion of rules too broad, too loose formulation,  rather simple formalization of how expert knowledge maybe added to NNs (9LgA)
> We do not agree with this point as it was our intention to provide a very general framework that can be used to incorporate expert knowledge in a flexible way.
> Indeed, the fact that "The presented theory is very general and simple" was also mentioned as a strength of the paper by the same reviewer. This is somewhat contradictory to the above statement.
> None of the reviewers have provided a clear argument why the notion of rules is too broad or too loose.
2. Examples do not go beyond what is possible by adding simple graph features to the node features (9LgA)
> We do not agree with this point as we have shown that our approach also outperforms those GNNs where the exact same information (features) that our model uses is provided as input to the GNNs, see the (features) results in Table 1.
> Moreover, our framework allows for message passing over arbitrary distances in the graph in one layer which is not possible by just adding simple graph features to the node features.
> In fact, we need only one message passing layer even for long range interactions in the graph which avoids over-smoothing and allows for a more efficient training.
> Indeed, this property should have been made more explicit in the paper.
3. Soundness of Theorem 1 (hyNs)
> We agree with this point that the statement of Theorem 1 is not fully proven in the appendix.
> Nevertheless, the proof to arbitrary convolutions is straightforward and we will provide a more detailed proof in a revised version of the paper.
4. Clarity of the introduction of rule based layers (hyNs), (kGau) and (yEHT)
> We agree with this point that the introduction of rule based layers could be made more clear.
> Nevertheless, two out of the four reviewers mention as strength that the presented theory is simple and clear.
5. Empirical results (9LgA), (kGau), (hyNs)
> We do not fully agree with the point that the empirical results are not convincing because they show that our approach outperforms GNNs in the fair comparison even if the same information (features) is provided to the models, see the (features) results in Table 1.
> Moreover, the experiments on the synthetic datasets show that the expressive power of our approach is higher than the expressive power of classical GNNs.
> We agree that it would be helpful to add the comparisons with more expressive architectures in Table 1 and Table 2. (hyNs)
> An additional theoretical result on the expressive power not provided in this work is that using adequate labeling functions, makes the model arbitrarily expressive. This is a very interesting point that we will add to the paper.

The comments on the strengths show that most of the reviewers agree that the idea of adding expert knowledge to NNs and GNNs using rule based layers is interesting, promising and novel.
We do not agree with the main concerns raised by the reviewer 9LgA that the notion of rules is too broad or too loose and that the examples do not go beyond what is possible by adding simple graph features to the node features as explained above and in the "Rebuttal by Authors" corresponding to the review 9LgA.
We agree with the reviewers that some points could be made more clear in the paper including the soundness of Theorem 1 and the minor issues raised by reviewer hyNs.
We will address these points in a revised version of the paper.

---

### Decision · Program_Chairs · 2024-09-25

**Decision:**

Reject

**Comment:**

The paper presents promising ideas concerning the integration of expert knowledge into GNNs. However, these ideas appear to be too preliminar and the review discussion highlighted a general (strong) concern regarding the fact that the approach relies heavily on the availability of almost oracular expert knowledge on the task, to be effective. This is a major factor playing against acceptance of the work, together with a yet undeveloped general framework sustaining the approach (e.g. lack of appropriate formalization of rules, of how rules can be composed, etc).